# ERK-mediated phosphorylation regulates SOX10 sumoylation and targets expression in mutant BRAF melanoma

Shujun Han[1], Yibo Ren[1], Wangxiao He[1], Huadong Liu[1], Zhe Zhi[1], Xinliang Zhu[1], Tielin Yang [1], Yu Rong[1], Bohan Ma[1], Timothy J. Purwin[2], Zhenlin Ouyang[1], Caixia Li[1], Xun Wang[1], Xueqiang Wang[1], Huizi Yang[1], Yan Zheng[3], Andrew E. Aplin[2,4], Jiankang Liu[1,5,6] & Yongping Shao[1]

In human mutant BRAF melanoma cells, the stemness transcription factor FOXD3 is rapidly induced by inhibition of ERK1/2 signaling and mediates adaptive resistance to RAF inhibitors. However, the mechanism underlying ERK signaling control of FOXD3 expression remains unknown. Here we show that SOX10 is both necessary and sufficient for RAF inhibitor-induced expression of FOXD3 in mutant BRAF melanoma cells. SOX10 activates the transcription of FOXD3 by binding to a regulatory element in FOXD3 promoter. Phosphorylation of SOX10 by ERK inhibits its transcription activity toward multiple target genes by interfering with the sumoylation of SOX10 at K55, which is essential for its transcription activity. Finally, depletion of SOX10 sensitizes mutant BRAF melanoma cells to RAF inhibitors in vitro and in vivo. Thus, our work discovers a novel phosphorylation-dependent regulatory mechanism of SOX10 transcription activity and completes an ERK1/2/SOX10/FOXD3/ERBB3 axis that mediates adaptive resistance to RAF inhibitors in mutant BRAF melanoma cells.

[1] Frontier Institute of Science and Technology, and Key Laboratory of Biomedical Information Engineering of Ministry of Education, School of Life Science and Technology, Xi'an Jiaotong University, Xi'an 710049, China. [2] Department of Cancer Biology, Thomas Jefferson University, Philadelphia, PA 19107, USA. [3] Department of Dermatology, the Second Affiliated Hospital, School of Medicine, Xi'an Jiaotong University, Xi'an 710004, China. [4] Sidney Kimmel Cancer Center, Thomas Jefferson University, Philadelphia, PA 19107, USA. [5] National & Local Joint Engineering Research Center of Biodiagnosis and Biotherapy, The Second Affiliated Hospital, Xi'an Jiaotong University, Xi'an 710004, China. [6] Tianjin Key Laboratory of Exercise Physiology and Sports Medicine, Tianjin University of Sport, Tianjin, China. Correspondence and requests for materials should be addressed to Y.S. (email: yongping.shao@mail.xjtu.edu.cn) or to J.L. (email: j.liu@mail.xjtu.edu.cn)

Small molecule inhibitors targeting BRAF and/or MEK kinases have achieved great success in the treatment of mutant BRAF melanoma[1–4]. However, clinical benefit of these agents is often limited by short-lived responses and acquired resistance via heterogeneous mechanisms[5, 6]. Since resistant tumor cells are derived from parental cells that survive the initial drug treatment[7], improving the initial treatment efficacy to maximally eliminate sensitive tumor cells may effectively delay the onset of durable acquired resistance. The initial responsiveness of mutant BRAF melanoma patients to RAF and/ or MEK inhibitors varies substantially and is influenced by tumor microenvironment and adaptive resistance[8–10]. Adaptive resistance involves a rapid and reversible rewiring of pro-survival signaling pathways in response to therapeutic agents[8]. Understanding the mechanisms of adaptive resistance will help to develop combinatorial therapeutic approaches that more efficiently eliminate tumor cells at the early treatment stage through synthetic lethal effects and prolong the progression-free survival.

In contrast to the highly diversified acquired resistance, only a few mechanisms of adaptive resistance to RAF inhibitors have been reported in melanoma, such as ERK1/2 reactivation, upregulation of RTKs and metabolic reprogramming[8]. One important example of adaptive resistance is the upregulation of the stem cell transcription factor, Forkhead box D3 (FOXD3) upon inhibition of ERK1/2 signaling in mutant BRAF melanoma cells[11, 12]. FOXD3 mediates adaptive resistance to RAF inhibitors by directly activating the expression of v-erb-b2 erythroblastic leukemia viral oncogene homolog 3 (ERBB3) at the transcriptional level and enhancing the responsiveness of melanoma cells to the ERBB3 ligand, neuregulin-1 (NRG1)[13]. Enhanced NRG1/ ERBB3 signaling activates the PI3K/AKT pathway and protects melanoma cells against the cytotoxic effect of RAF inhibitors. Although the role of FOXD3 as a mediator of adaptive resistance to RAF inhibitors in mutant BRAF melanoma cells has been well established, how ERK signaling controls FOXD3 expression remains unclear.

Sex determining region Y (SRY) related HMG box-containing factor 10 (SOX10) is a member of the SOX family transcription factors that plays pivotal regulatory roles in the development of neural crest and the melanocyte lineage. SOX10 haploinsufficiency causes pigmentation defects and Waardenburg syndromes in human[14, 15]. SOX10 regulates the proliferation, survival and melanogenesis of melanocytes by activating its target genes including *Mitf*, *Dct*, *Tyr*, and *Tyrp1*[14]. SOX10 is also important for the initiation and maintenance of melanoma[16] and promotes the migration and invasion of melanoma cells[17]. In a heterogeneous melanoma cell population, cells with low-SOX10 expression are associated with increased TGF-β signaling and elevated EGFR/PDGFR expression, which leads to a reversible adaptive resistance to RAF inhibitors[18]. Recently, SOX10 was found to regulate the expression of the long non-coding RNA (lncRNA) SAMMSON, which is expressed in 90% of human melanoma and plays an oncogenic role[19].

While the importance of SOX10 in embryonic development and melanoma progression has been well recognized, the regulation of SOX10 remains poorly characterized. SOX10 transcription has been shown to be controlled by multiple species-conserved regulatory sequences in the upstream region and binding sites of a variety of transcriptional factors have been discovered in these sequences[20]. Post-translational modifications also participate in the regulation of SOX10. For example, sumoylation of SOX10 regulates its transcriptional activity[21, 22] and FBXW7-mediated ubiquitination of SOX10 controls its protein stability[23].

In this study, we identify SOX10 as a transcriptional activator of FOXD3 downstream of ERK1/2 signaling. SOX10 activates the transcription of FOXD3 by direct binding to a regulatory sequence in the promoter of FOXD3. We further show that ERK phosphorylates SOX10 at T240 and T244, which inhibits the sumoylation of SOX10 at K55 and subsequently the transcription activity toward its target genes. Our findings not only delineate a signaling network that governs the FOXD3-mediated adaptive resistance to RAF inhibitors in mutant BRAF melanoma but also demonstrate an intricate regulatory switch of SOX10 transcription activity that involves interplay between phosphorylation and sumoylation.

## Results

**SOX10 is necessary and sufficient for FOXD3 induction.** Blocking ERK signaling in mutant BRAF melanoma cells with RAF or MEK inhibitors induces FOXD3 expression at the transcriptional level[11]; however, the underlining mechanism of this regulation is unknown. Studies have shown that FOXD3 and SOX10 are two transcription factors that are both expressed in pre-migratory neural crest and play similar regulatory roles in the development of neural crest[24, 25]. We analyzed whether there is a regulatory relationship between SOX10 and FOXD3 in melanoma cells. We first evaluated the correlation between expression of SOX10 and FOXD3 in melanoma patients based on two independent data sets: RNA-seq data from the TCGA research network (http://cancergenome.nih.gov) and microarray data from a study by Talantov et al.[26]. Spearman correlation analysis successfully detected a positive correlation of SOX10 with several of its known targets including MITF, DCT, and TYR, confirming previous findings and the validity of our analysis. Notably, a positive correlation was also found between SOX10 and FOXD3 in both data sets when analyzing all melanoma genotypes and selectively BRAF mutant melanoma (Supplementary Table 1).

We then investigated whether SOX10 is a mediator of the ERK-dependent regulation of FOXD3 in mutant BRAF melanoma cells. SOX10 expression was depleted using two independent SOX10-specific siRNAs in mutant BRAF melanoma cells which were then treated with the RAF inhibitor, Vemurafenib, for various times. Consistent with previous studies, Vemurafenib treatment resulted in a rapid and time-dependent induction of FOXD3 at both protein and mRNA levels (Fig. 1a, b). SOX10 knockdown using either of the siRNAs effectively reduced Vemurafenib-induced FOXD3 levels at both mRNA and protein levels, indicating that SOX10 is required for FOXD3 induction associated with inhibition of ERK signaling (Fig. 1a, b). This SOX10-dependent induction of FOXD3 by inhibition of ERK1/ 2 signaling is durable for at least 120 h (Supplementary Figs. 1 and 16) and is also present in melanoma cells treated with a combination of RAF and MEK inhibitors (Supplementary Figs. 2, 17, 18). In addition, the ERK1/2/SOX10/FOXD3 axis appears to be specific to mutant BRAF melanoma cells since N-RAS mutant melanoma cells have no detectable level of basal or induced FOXD3 expression (Supplementary Figs. 3 and 19). To rule out the potential off-target effects of siRNAs, we confirmed the regulation of FOXD3 by SOX10 by a rescue experiment, in which the endogenous SOX10 was ablated by RNA interference while an exogenous HA-tagged and siRNA-resistant SOX10 cDNA was introduced through a lentiviral system. Two Tet repressor-expressing mutant BRAF cell lines, A375-TR and 1205Lu-TR were used to transduce the lentivirus so that the expression of the exogenous HA-SOX10 is controllable by doxycycline[13]. As expected, doxycycline induced the expression of the exogenous HA-SOX10 in A375-TR and 1205Lu-TR cells and the expression was resistant to SOX10 siRNAs (Fig. 1c). In the absence of doxycycline, FOXD3 was induced by Vemurafenib treatment but ablated upon SOX10 depletion as previously seen (Fig. 1c). Of

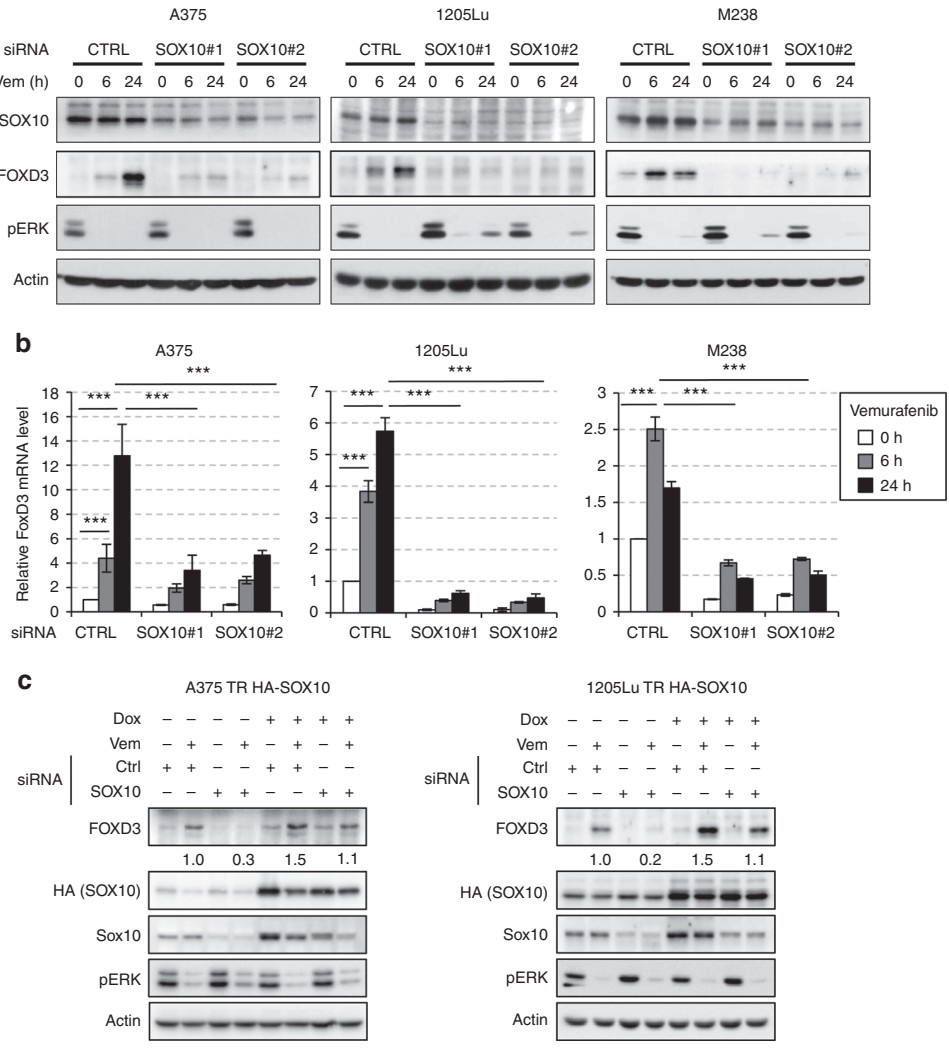

**Fig. 1** SOX10 is necessary and sufficient for FOXD3 induction by ERK signaling inhibition. **a** Melanoma cells were transfected with non-targeting control or SOX10-specific siRNAs for 72 h and treated with 2 μM Vemurafenib for 0, 6, and 24 h before being lysed for western blot analysis. **b** Same as (**a**) except that after siRNA transfection and Vemurafenib treatment, cells were collected to isolate total RNA for qRT-PCR analysis on FOXD3 using actin as the internal control. Average results from three independent experiments are shown. Error bars represent standard deviation. Significance was determined by ANOVA one-way test, ***$p < 0.001$. **c** 1205Lu-TR HA-SOX10 and A375-TR HA-SOX10 cells were transfected with control or SOX10 siRNAs for 72 h in the presence or absence of 100 ng mL$^{-1}$ Doxycycline. The cells were then treated with 2 μM Vemurafenib for 24 h and lysed for western blot analysis. Quantitations of FOXD3 expression based on band intensity are shown below the blots. Uncropped images are shown in Supplementary Fig. 9

note, expression of exogenous HA-SOX10 not only enhanced the inhibitor-induced FOXD3 level, but also completely rescued FOXD3 induction by Vemurafenib when endogenous SOX10 was depleted. Together, our loss-of-function and rescue experiments consistently demonstrate that SOX10 is a necessary and sufficient transcription activator of FOXD3 downstream the ERK signaling.

**SOX10 directly regulates FOXD3 transcription.** Since the regulation of FOXD3 expression by SOX10 occurred at the mRNA level, we wondered whether SOX10 can directly regulate FOXD3 promoter activity. To test this, dual luciferase reporter assays were performed using a 1.6 kb region of the human FOXD3 promoter linked to luciferase gene and an increasing amount of SOX10-expressing plasmids in HEK293T cells. SOX10 overexpression enhanced FOXD3 promoter activity in a dose-dependent manner (Fig. 2a). SOX10 binds to a consensus DNA sequence 5′-[A/T][A/T]CAA[A/T]G-3′[27] and bioinformatics analysis of the 1.6 kb FOXD3 promoter fragment uncovered three putative SOX10

binding sites upstream the transcription starting site (Fig. 2b). To determine which putative site accounts for the transactivation activity of SOX10, we individually mutated the three sites and examined the impact on FOXD3 promoter activity. Over-expression of SOX10 enhanced wild-type (WT) FOXD3 promoter activity by 50% (Fig. 2c). Mutation of site 1 or site 2 had negligible effects on SOX10-enhanced promoter activity, while disruption of site 3 completely abolished the transactivation activity of SOX10, indicating that site 3 is essential for the transactivation function of SOX10 toward FOXD3 promoter. Interestingly, site 3, but not site 1 or 2 was evolutionary conserved among different species, further supporting it being an important regulatory element (Fig. 2d). To investigate whether SOX10 directly binds to FOXD3 promoter in vivo, ChIP analysis was performed using the HA antibody (for exogenous SOX10) and a primer set spanning site 3. As expected, no enrichment was detected in HA versus IgG immunoprecipitates on a genomic region between the GAPDH and CNAP1 genes, which served as a negative control. By

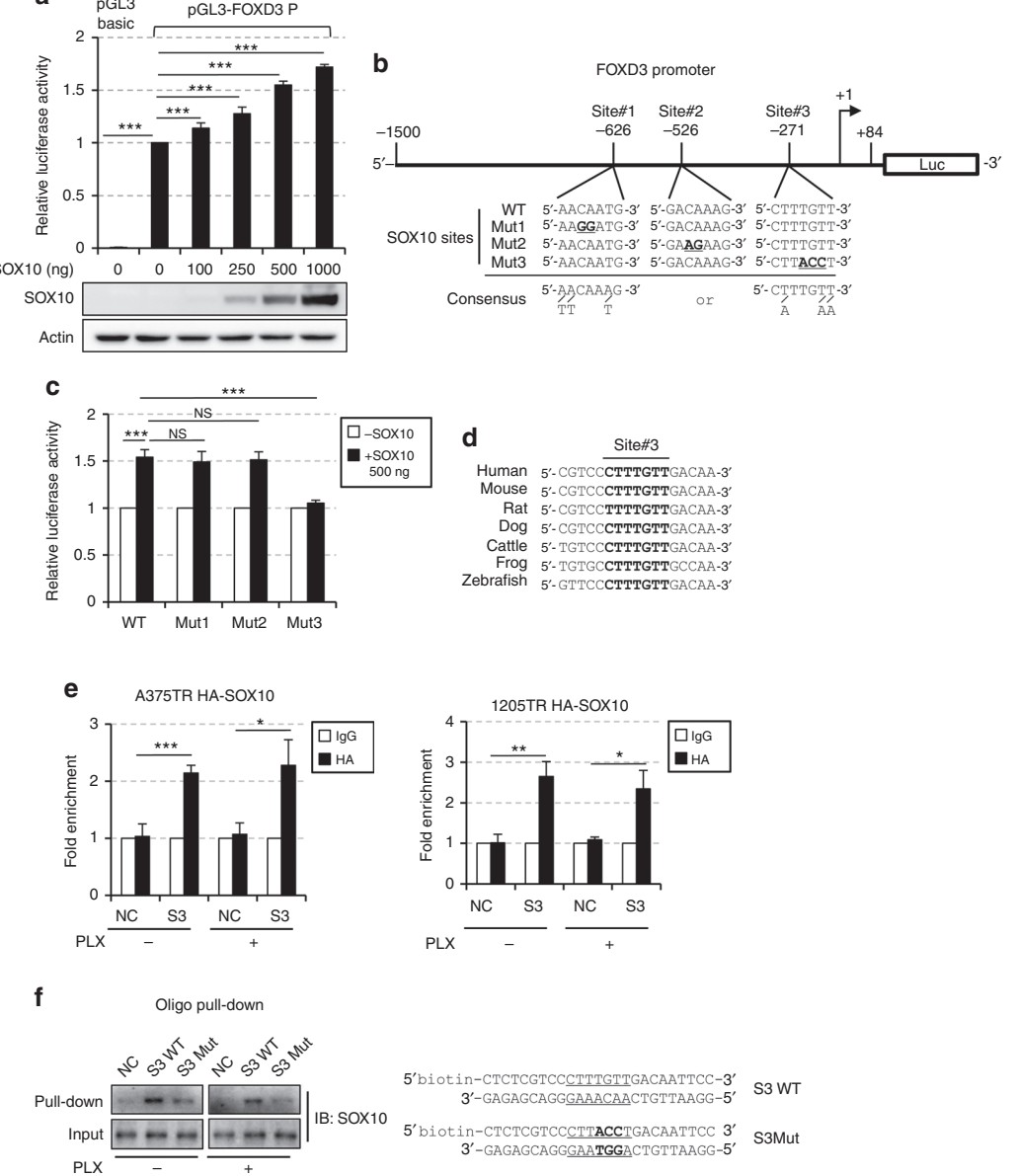

**Fig. 2** SOX10 activates the transcription of FOXD3 by direct binding to FOXD3 promoter. **a** HEK293T cells were co-transfected with 500 ng pGL3-FOXD3 (or pGL3-Basic as negative control), 50 ng pRL-TK and an increasing amount of pLentipuro/TO/HA-SOX10 plasmids. After 48 h, cells were lysed and dual-luciferase assays were performed. Average relative luciferase activities from three experiments are shown. The expression of SOX10 was verified by western blot. Error bars represent standard deviation. Significance was determined by ANOVA one-way test, ***$p < 0.001$. **b** A schematic illustration of FOXD3 promoter region. The positions and sequences of three putative SOX10 binding sites were highlighted. The +1 arrow designated the transcription initiation site. The mutated SOX10 binding sites and the consensus motif are shown. Mutated nucleotides were underlined. **c** HEK293T cells were co-transfected with 500 ng pGL3-FOXD3 promoter constructs carrying either WT sequence or mutations in either of the three putative SOX10 binding sites, 50 ng pRL-TK and 500 ng pLentipuro/TO/HA-SOX10 for 48 h. Cells were then lysed for dual-luciferase assay. Average relative luciferase activities from three experiments are shown. Error bars represent standard deviation. Significance was determined by ANOVA one-way test, ***$p < 0.001$. **d** Sequence alignment of SOX10 binding site #3 in FOXD3 promoters from different species. The SOX10 binding sites were in bold. **e** A375-TR HA-SOX10 and 1205Lu-TR HA-SOX10 cells were treated with 2 μM Vemurafenib for 6 h. Occupancy of SOX10 (HA) on a region surrounding site #3 in FOXD3 promoter and a region between the GAPDH and CNAP1 genes (negative control) was evaluated by ChIP analysis. Average results from three independent experiments are shown. Error bars represent standard deviation. Significance was determined by ANOVA one-way test, *$p < 0.05$; **$p < 0.01$; ***$p < 0.001$. **f** Oligo pull-down assays were performed using nuclear extracts from A375 cells treated with or without 2 μM Vemurafenib for 6 h and biotin-labeled FOXD3 promoter fragments containing WT or mutated SOX10 binding site #3. Non-biotinylated DNA fragments (NC) were used as a negative control. The nucleotide sequences of promoter fragments are shown on the right. SOX10 binding sites were underlined and mutated nucleotides were highlighted in bold. Uncropped images are shown in Supplementary Fig. 10

contrast, the site 3 region of FOXD3 promoter was significantly enriched in HA immunoprecipitates versus the IgG control (Fig. 2e). Vemurafenib treatment did not alter the level of enrichment at site 3, indicating ERK inhibition does not affect the chromatin occupancy by SOX10 at the FOXD3 promoter. We next performed oligonucleotide pull-down assays to interrogate the direct interaction between SOX10 and site 3. A 25-bp biotinylated FOXD3 promoter fragments containing site 3 efficiently pulled down SOX10 from the nuclear extract of A375 cells (Fig. 2f). However, the amount of SOX10 pulled down was reduced when site 3 was mutated in the same promoter fragment. In addition, Vemurafenib treatment had marginal effects on the efficiency of SOX10 pull-down, which was consistent with the ChIP results. Therefore, we conclude that SOX10 likely activates FOXD3 transcription by direct binding to a −271 regulatory element in the FOXD3 promoter region.

**ERK2 phosphorylates SOX10 at T240 and T244**. We next determined how ERK signaling regulates the transcription activity of SOX10 toward FOXD3. FOXD3 induction by ERK inhibition is independent of increased SOX10 protein level (Fig. 1) and altered nuclear localization (Supplementary Fig. 4). Furthermore, ERK inhibition by Vemurafenib does not seem to affect the binding of SOX10 to FOXD3 promoter (Fig. 2e, f). These observations, together with the rapid induction rate of FOXD3 within hours of ERK inhibition (Fig. 1a, b) suggested the notion that ERK signaling may regulate the transcriptional activity of SOX10 via post-translational modification. Guided by the ERK consensus phosphorylation motif "pxT/Sp", we identified two putative ERK phosphorylation sites, T240 and T244 in SOX10. Interestingly, these two sites are highly conserved among species and the SOXE family proteins (Fig. 3a). Phosphorylation of two corresponding sites in SOX9 (T236 and T239) was detected in breast cancer cells[28]. Furthermore, a previous proteomic study detected phosphorylated SOX10 tryptic peptides (residue 216–246) harboring the two putative ERK sites (T240 and T244) in a mutant BRAF melanoma cell line although the exact phosphorylation sites were not determined[29]. Based on these observations, we tested whether SOX10 is phosphorylated in vivo at T240 and/or T244. Four tryptic peptides of SOX10 (spanning residue 216–246) that carry either none, single or double phosphorylation sites were individually synthesized (Supplementary Fig. 5) and used as peptide standards in a multiple reactions monitoring (MRM) mass spectrometry analysis on HA-SOX10 immunoprecipitated from A375-TR HA-SOX10 cell lysates. As expected, phosphorylation of T240 or T244, and both sites together was detected from A375 melanoma cell lysates (Fig. 3b). Importantly, treatment of Vemurafenib reduced the levels of SOX10 phosphorylation at both single and double sites (Fig. 3c).

To further examine whether ERK2 can directly phosphorylate SOX10 at T240 and/or T244, In vitro kinase assays were performed using recombinant activated ERK2 kinase and synthetic SOX10 peptides (236-HGPPTPPTTPKTELQ-250) with WT sequence or alanine replacement at T240 and/or T244. The reaction products were analyzed by LC–Mass Spectrometry. Three peaks were detected for the WT peptides, which corresponded to unphosphorylated (MW: 1600D), single-phosphorylated (MW: 1680D) and double-phosphorylated (MW: 1760D) species respectively (Fig. 3d, Supplementary Fig. 6). For T240A or T244A SOX10 peptides, only unmodified and single-phosphorylated species were detected. However, no phosphorylation was detected with the AA peptides. Together, these results indicate that ERK2 can directly phosphorylate SOX10 at T240 and/or T244 residues. To further validate the phosphorylation of SOX10 by ERK kinases in vivo, we

individually immunoprecipitated WT, T240A, T244A, and AA HA-SOX10 variants from lentivirus transduced A375 cells treated with or without the ERK inhibitor SCH772984 and probed phospho-threonine using an antibody targeting the PXpTP motif. As expected, phospho-threonine was successfully detected in WT HA-SOX10 and the signal was reduced when cells were treated with SCH772984 (Fig. 3e). Importantly, significantly less or no phospho-threonine signals were detected in T240A, T244A, or AA HA-SOX10, confirming the phosphorylation of T240 and T244 sites by ERK kinases in vivo. Moreover, the phosphorylation of SOX10 at T240/T244 was also observed in 293T cells and was inhibited by MEK inhibitor (Fig. 3f), indicating these modifications are not cellular context specific.

**Phosphorylation of SOX10 inhibits its transcription activity**. Following the discovery of the two ERK phosphorylation sites in SOX10, we next asked whether T240 and/or T244 phosphorylation regulates the transcriptional activity of SOX10 toward FOXD3. Endogenous SOX10 was depleted in A375-TR or 1205Lu-TR cells and siRNA-resistant, phosphomimetic variants of HA-SOX10 including T240E, T244E, or EE were ectopically expressed through lentiviral vectors so that the transcription activity of SOX10 variants toward FOXD3 can be compared without the interference of endogenous WT SOX10. As shown in Fig. 1c, expression of WT HA-SOX10 efficiently rescued FOXD3 induction by ERK inhibition in melanoma cells depleted of endogenous SOX10. By contrast, in the absence of endogenous SOX10, the phosphomimetic T240E or T244E replacement inhibited the induction of FOXD3 by Vemurafenib and the EE mutation almost completely blocked the induction of FOXD3 (Fig. 4a, b), suggesting that T240 and/or T244 phosphorylation compromise the transcription activity of SOX10 toward FOXD3. Importantly, we found that the phosphorylation-dependent regulation of SOX10 transcription activity is common toward other SOX10 targets including MITF, TYR, and SAMMSON. Depletion of endogenous SOX10 caused reduced expression of these target genes which was fully rescuable by expression of exogenous WT but not EE SOX10 (Fig. 4c). It is worth noting that all ectopic SOX10 mutants were expressed at a comparable level to the endogenous SOX10 and to WT HA-SOX10. Therefore, the reduced capacities of SOX10 phosphomimetic mutants to activate FOXD3 expression are not due to inefficient expression but more likely caused by impaired transcriptional activity. We also noticed that expression of exogenous SOX10 variants, regardless of their mutational status, all enhanced the expression levels of SOX10 targets in the presence of endogenous SOX10 and Vemurafenib. While the detailed mechanism is still unknown, one possible explanation is that expression of exogenous SOX10 relieves the inhibiting effects on the transcription activity of endogenous SOX10 by titrating out the inhibitory factors. Nevertheless, our knockdown/re-expression experiments clearly indicated that T240 and/or T244 phosphorylation inhibits the transcription activity of SOX10.

**Sumoylation is required for SOX10 transcriptional activity**. Sumoylation regulates SOXE protein transcriptional activity and function in early development of neural crest and ear[22]. SOX10 contains two sumoylation motifs (K55 and K357), which are conserved among different species and in its family member, SOX9 (Fig. 5a). To scrutinize the sumoylation of SOX10 at these two sites, Flag-tagged SUMO1 and HA-tagged SOX10 variants, WT, K55R, K357R, and 2KR, were co-expressed in HEK293T cells and the lysates were analyzed by western blot. In addition to the unmodified HA-SOX10 band at around 65 KD, a higher molecular weight band (above 100 KD) was observed for

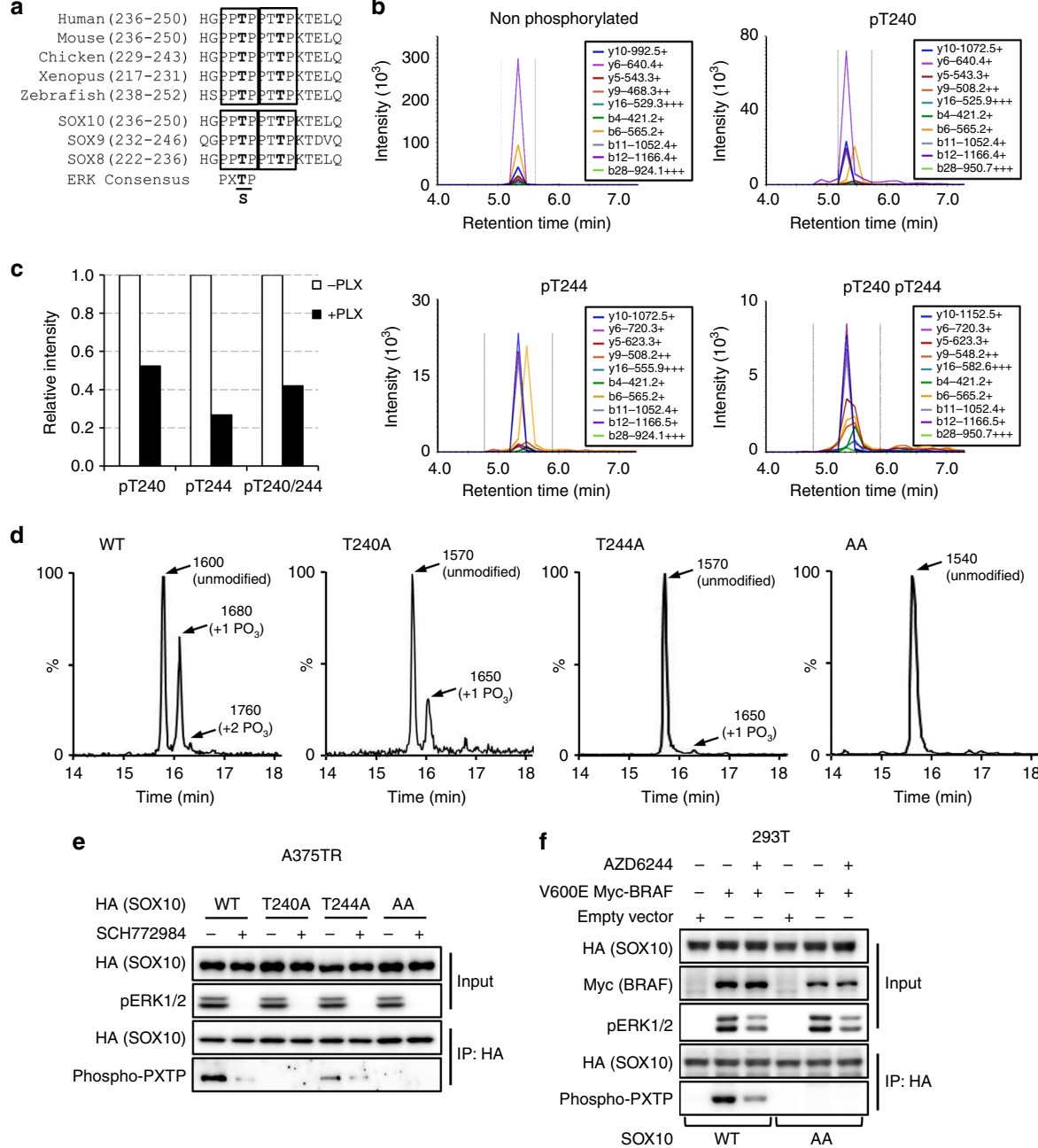

**Fig. 3** ERK2 directly phosphorylates SOX10 at T240 and T244. **a** Sequence alignment of the putative ERK phosphorylation motifs in SOX10 from different species (top) and among SOXE family proteins (bottom). The putative phosphorylation motifs were highlighted in box and phosphorylation sites (T) are shown in bold. The consensus ERK phosphorylation motif is shown below the sequences. **b** In vivo detection of SOX10 phosphorylation at T240 and T244. HA-SOX10 proteins were immunoprecipitated from A375-TR HA-SOX10 cell lysates, digested by trypsin and analyzed by multiple reactions monitoring (MRM) mass spectrometry. MRM spectra of non-phosphorylated and T240/T244-phosphorylated SOX10 tryptic fragments are shown. **c** Quantitation of SOX10 phosphorylation in A375 cells treated with or without 2 μM Vemurafenib for 6 h. The areas of the peptide peaks in MRM chromatograms were measured to estimate the relative quantities of corresponding peptides. A SOX10 tryptic fragment (183 AAQGEAECPGGEAEQGGTAAIQAHYK 208, 848.7188+++) was used as the internal control. **d** In vitro kinase assays were performed using recombinant ERK2 (activated) and synthetic SOX10 peptides (236-HGPPTPPTTPKTELQ-250) and the reaction products were analyzed by LC–MS. The HPLC results for SOX10 peptide variants including WT, T240A, T244A, and AA are shown. Different peptide species detected in the reaction products were designated by arrows with their molecular weights and identities shown beside. Mass spectrometry results were included in the supplemental information (Supplementary Fig. 3). **e** WT, T240A, T244A, AA HA-SOX10 proteins were precipitated from lysates of lentivirus-transduced A375 cells treated with or without 1 μM SCH772984 and probed with anti-HA or anti-phospho-PXTP antibodies. Uncropped images are shown in Supplementary Fig. 10. **f** 293T cells were transduced with WT or AA HA-SOX10 constructs along with empty vector or V600E BRAF constructs, ±AZD6244 treatment. Cells were lysed and HA-SOX10 were immunoprecipitated and probed with indicated antibodies. Uncropped images are shown in Supplementary Fig. 11

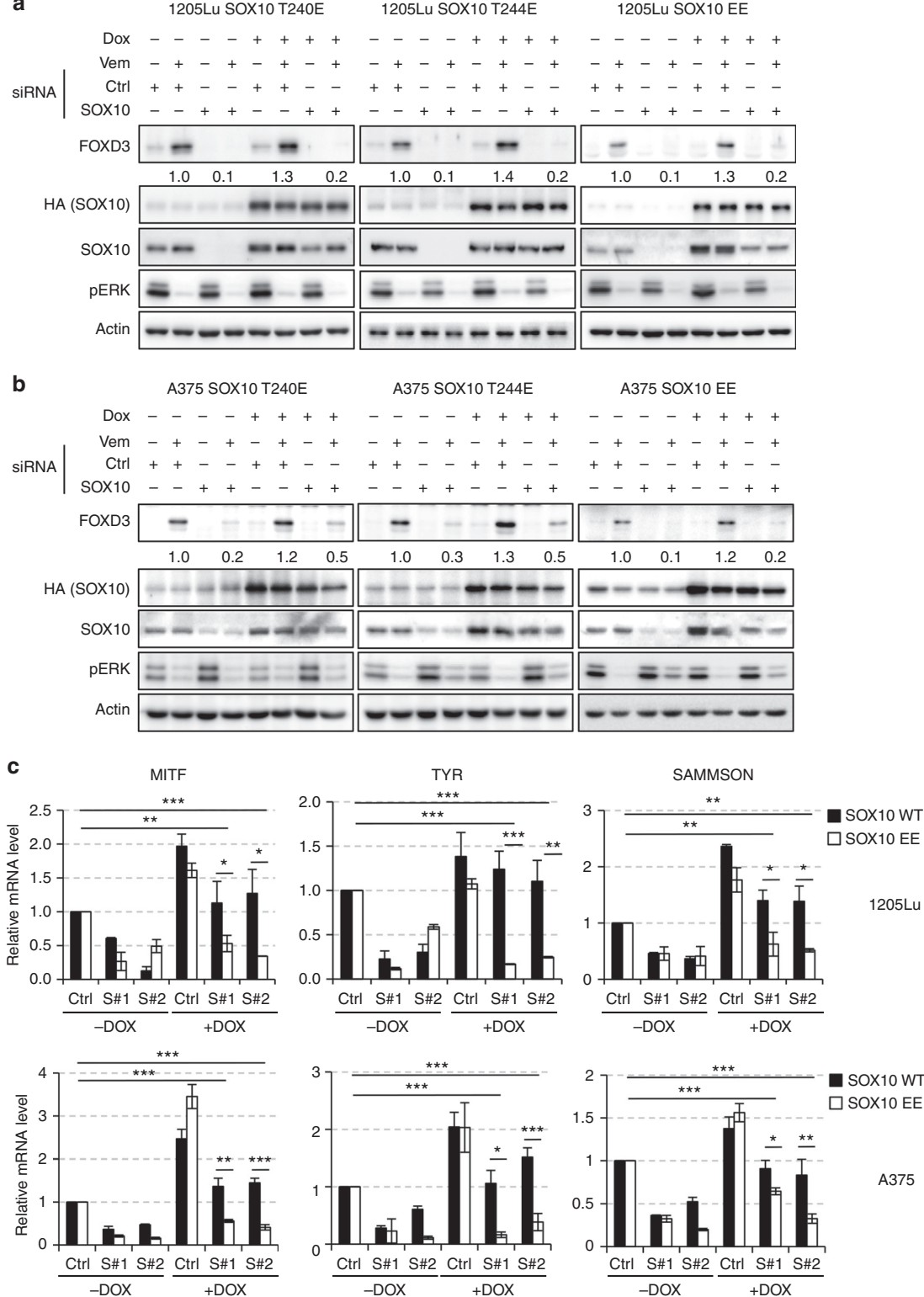

**Fig. 4** Phosphorylation at T240 and/or T244 inhibits transcriptional activity of SOX10 toward FOXD3 and other targets. **a** 1205Lu-TR cells transduced with lentiviruses carrying siRNA-resistant, HA-tagged SOX10 cDNA variants including T240E, T244E, and EE were transfected with SOX10 siRNAs for 72 h in the absence or presence of 100 ng ml$^{-1}$ doxycycline. Cells were then lysed and analyzed by western blot on indicated proteins. Actin was used as a loading control. Quantitation of FOXD3 expression is shown below the corresponding blots. **b** Same as (**a**) except that A375-TR cells were used. **c** 1205Lu-TR (top) or A375-TR (bottom) cells transduced with either WT or EE HA-SOX10 lentiviruses were transfected with SOX10 siRNAs for 72 h with or without 100 ng ml$^{-1}$ doxycycline. Cells were then treated with 2 μM Vemurafenib for 24 h and lysed for total RNA isolation and qRT-PCR analysis. Average results from three independent experiments are shown. Error bars represent standard deviation. Significance was determined by ANOVA one-way test,*$p < 0.05$; **$p < 0.01$; ***$p < 0.001$. Uncropped images are shown in Supplementary Fig. 12

WT and K357R HA-SOX10, but not for K55R or 2KR mutant (Fig. 5b). To validate the identity of the higher molecular weight band, HA-SOX10 immunoprecipitates were probed with anti-Flag antibody for SUMO1 detection. Similar to the results of anti-HA blots, Flag antibody detected a band above 100 KD in WT and K357R immunoprecipitates but not in K55R or 2KR mutant (Fig. 5b), indicating that the higher molecular weight band indeed represents the sumoylated SOX10 and that SOX10 is sumoylated

at K55. To investigate whether K55 sumoylation modulates SOX10 transcription activity toward FOXD3, we ectopically expressed K55R, K357R, or 2KR HA-SOX10 mutants in 1205Lu-TR or A375-TR cells depleted of endogenous SOX10 and monitored the impact on FOXD3 induction by Vemurafenib. In accordance with the sumoylation status at K55 and K357, both the K55R and 2KR mutants had severely impaired abilities to activate FOXD3 transcription upon ERK signaling inhibition

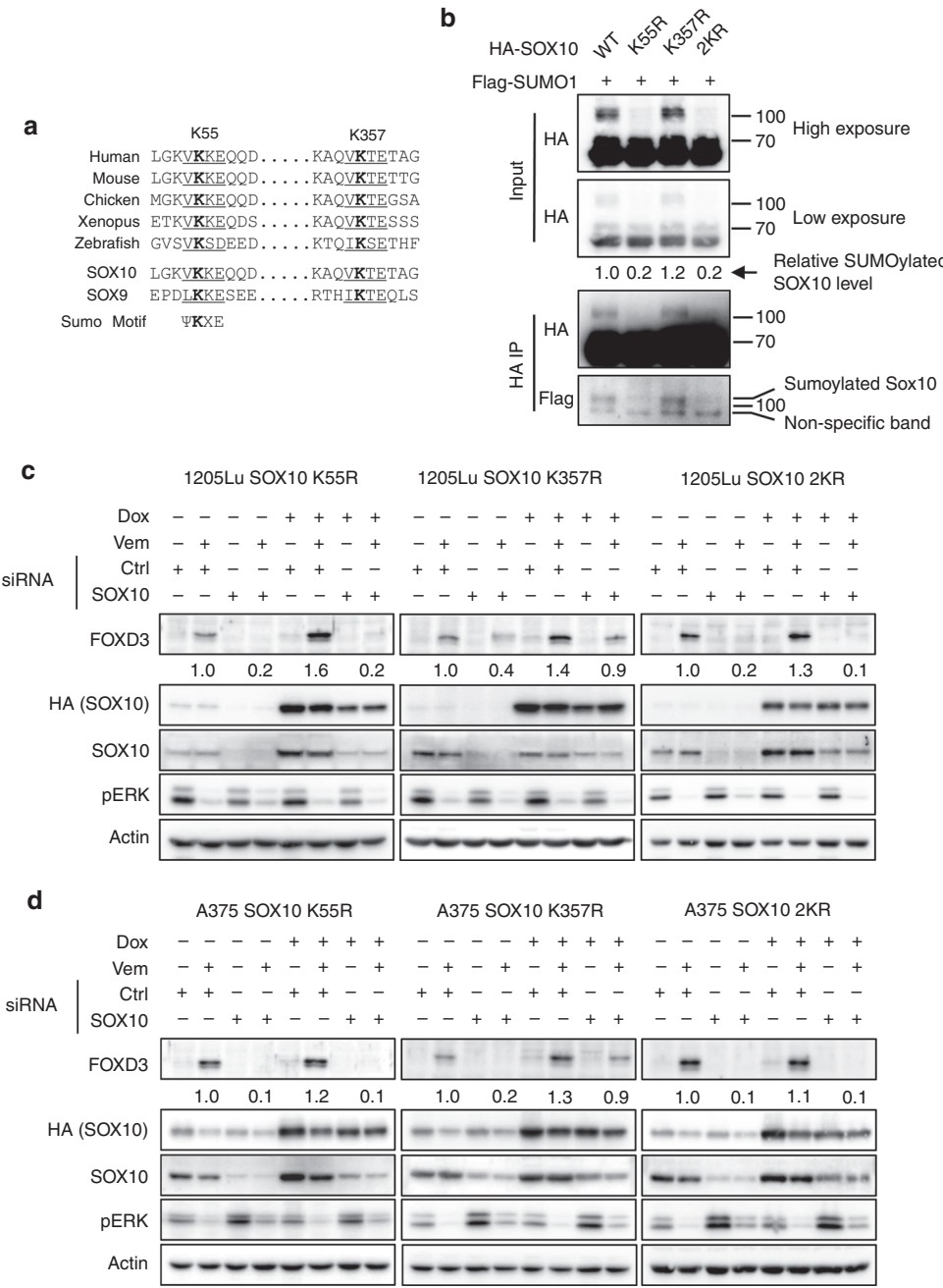

**Fig. 5** K55 sumoylation is essential for the transcriptional activity of SOX10 toward FOXD3. **a** Alignment of sumoylation motifs in SOX10 from different species and SOX9. Putative sumoylation sites were underlined and consensus sumoylation motif is shown. **b** HEK293T cells were co-transfected with plasmids expressing Flag-SUMO1 and one of the HA-SOX10 variants including WT, K55R, K357R, and 2KR. After 48 h, immunoprecipitation was performed with HA-tag antibody. Inputs and immunoprecipitates were analyzed by western blot. Relative levels of SOX10 sumoylation were quantitated by normalizing the intensities of the sumoylated bands against the non-sumoylated bands. **c** 1205Lu-TR cells transduced with lentiviruses carrying siRNA-resistant HA-SOX10 cDNA variants including K55R, K357R, and 2KR were transfected with SOX10 siRNAs for 72 h in the absence or presence of 100 ng ml$^{-1}$ doxycycline. Cells were then lysed and analyzed by western blot on indicated proteins. Actin was used as a loading control. Quantitation of FOXD3 expression is shown below the corresponding blots. **d** Same as (**b**) except that A375-TR cells were used. Uncropped images are shown in Supplementary Fig. 13

while loss of K357 had negligible effect (Fig. 5c, d). These results demonstrated that SOX10 is sumoylated at K55 and this modification is important for the transcriptional activity of SOX10 toward FOXD3.

**Phosphorylation interferes with the sumoylation of SOX10**. In light of the similar functional defects of the phosphomimetic mutants (T240E, T244E, EE) and Sumo-disrupting mutants (K55R, 2KR) of SOX10, we hypothesized that there may be interplay between these two post-translational modifications. To test this, we comparatively analyzed the sumoylation status of WT and phosphomimetic mutants (T240E, T244E, and EE) of SOX10. As shown in Fig. 6a, T240E or T244E SOX10 had decreased levels of sumoylation compared with WT SOX10 and the EE mutation reduced SOX10 sumoylation even further. These observations were well correlated with results from prior functional studies on phosphomimetic (Fig. 4) and sumo-defective SOX10 mutants (Fig. 5c, d) and supported a notion that phosphorylation at T240 and/or T244 inhibits the sumoylation of SOX10, thus inactivating SOX10 for FOXD3 transcription. To elucidate how phosphorylation of SOX10 may inhibit its sumoylation, we examined the interaction of WT or EE SOX10 with the sumo E2 ligase UBC9, an essential component of the sumoylation machinery. Reciprocal immunoprecipitation reliably detected the interaction between SOX10 and UBC9 (Fig. 6a), which was in accordance with previous reports[22]. Importantly, the SOX10/UBC9 interaction was weakened by the phosphomimetic EE mutation (Fig. 6b) and knockdown of UBC9 diminished the sumoylation of WT SOX10 (Fig. 6c). We then performed GST-pull-down assay to further verify the interaction between SOX10 and UBC9. As shown in Fig. 6d, WT SOX10 was efficiently pulled down by recombinant GST-UBC9 but not by the GST control, confirming the interaction between the SOX10 and UBC9. Moreover, EE SOX10 was pulled down less by GST-UBC9 when compared with WT SOX10. The above results indicated that phosphorylation of SOX10 at T240/T244 may inhibit SOX10 sumoylation at least partly by interfering with the interaction between SOX10 and UBC9.

To further confirm the role of SOX10 sumoylation in FOXD3 activation and the interplay between SOX10 phosphorylation and sumoylation, we performed dual-luciferase assays using the FOXD3 promoter reporter and a panel of HA-SOX10 variants including the sumoylation site mutants (K55R, K357R, and 2KR HA-SOX10), phosphomimetic mutant (EE HA-SOX10) and non-sumoylatable phosphomimetic mutant (2KR/EE HA-SOX10). Taylor et al. have demonstrated that the C-terminal SUMO1 fusion of SOX10 successfully recapitulated the function of sumoylated SOX10 at K357, a site close to the C-terminal end (21). In our system, we found that sumoylation of the N-terminal site K55 was more important than K357 for SOX10's transcription activity on FOXD3. Therefore, we additionally included two phosphomimetic mutants that are either N-terminally or C-terminally fused to SUMO1 to mimick constitutive sumoylated SOX10 (C-SUMO1/EE, EE HA-SOX10 with C-terminal SUMO1 fusion and N-SUMO1/EE, EE HA-SOX10 with N-terminal SUMO1 fusion). In accordance with our western blot results, K55R and 2KR SOX10 failed to activate FOXD3 promoter while K357R SOX10 retained WT activity (Figs 5c, d, 6e). The phosphomimetic mutants EE and 2KR/EE SOX10 lost their activities, confirming that T240/T244 phosphorylation compromises the transcription activity of SOX10 on FOXD3 (Figs 4a, b, 6e). Interestingly, we found that the N-terminal but not C-terminal SUMO1 fusion restored the transcription activity of EE SOX10 on FOXD3 promoter (Fig. 6e), supporting the idea that

phosphorylation of SOX10 may regulate its transcription activity through altering SOX10 sumoylation at K55.

**SOX10 depletion sensitizes melanoma cells to RAFi**. Since FOXD3 induction in melanoma cells following RAF inhibitor treatment promotes adaptive resistance by upregulating ERBB3 and activating the NRG1/ERBB3/AKT signaling[13], we hypothesized that depletion of SOX10, the upstream regulator of FOXD3, would block the FOXD3/ERBB3/AKT axis and sensitize melanoma cells to the RAF inhibitors. To test this, we evaluated the ERBB3/AKT signaling and vemurafenib-induced apoptosis in melanoma cells depleted of SOX10. Consistent with previous reports, vemurafenib treatment enhanced ERBB3 expression and improved the sensitivity of melanoma cells to ERBB3 ligand, NRG1, as assessed by the phosphorylation of AKT (Fig. 7a). Importantly, SOX10 knockdown almost completely blocked the upregulation of ERBB3 and the NRG1-dependent activation of AKT signaling by vemurafenib. Similar results were observed in melanoma cells treated with a combination of RAF and MEK inhibitors and for extended periods of time (Supplementary Fig. 2). These results suggested that SOX10 can modulate the activity of NRG1/ERBB3/AKT pathway by controlling FOXD3 expression. The impact of SOX10 depletion on melanoma cell proliferation and apoptosis was then analyzed by MTT assay and annexin V/PI staining respectively. SOX10 knockdown alone inhibited melanoma cell growth (Supplementary Fig. 7) but only had a moderate influence on apoptosis (Fig. 7b). However, when combined with the RAF inhibitor, SOX10 depletion increased RAF inhibitor-induced apoptosis rate from 25 to 48% in 1205Lu cells and from 15 to 27% in A375 cells, respectively (Fig. 7b, Supplementary Fig. 8). In the mouse xenograft model, SOX10 knockdown also blocked FOXD3/ERBB3 induction by Vemurafenib, reduced tumor growth and further enhanced the tumor-inhibiting capacity of Vemurafenib (Fig. 7c, d). Therefore, SOX10 depletion can sensitize mutant BRAF melanoma cells to Vemurafenib in vitro and in vivo.

## Discussion

Melanoma cells may elicit an adaptive resistance which rapidly activates the survival signals to protect against the cytotoxic effects of RAF inhibitors until acquired resistance takes over. One important mediator of adaptive resistance in mutant BRAF melanoma cells is the lineage-specific transcription factor, FOXD3, which undergoes rapid transcriptional induction upon inhibition of ERK1/2 signaling and activates the ERBB3/PI3K/AKT pathway[13]. Mechanistically, how FOXD3 expression is induced by ERK inhibition remains unknown. In this study, we discover SOX10 as a transcription activator of FOXD3 downstream of the ERK1/2 signaling. We show that SOX10 activates FOXD3 transcription through binding to a regulatory site in the promoter region and that ERK directly phosphorylates SOX10 at T240 and T244, which inhibits sumoylation of SOX10 at K55 and consequently the transcriptional activity of SOX10 that is dependent on this modification. Our work completes an ERK/SOX10/FOXD3/ERBB3 pathway that governs the FOXD3-mediated adaptive resistance to RAF/MEK inhibitors in mutant BRAF melanoma. It also describes a novel regulatory mechanism of SOX10 transcriptional activity that involves interplay between two post-translational modification events: phosphorylation and sumoylation.

Previous works have shown that two conserved distal enhancer elements, NC1 and NC2, participate in the regulation of FOXD3 transcription by interacting with multiple transcription factors, such as Pax7, Msx1/2, Ets1, and Zic1[30]. In addition to these distal enhancer elements, high level of sequence conservation was also

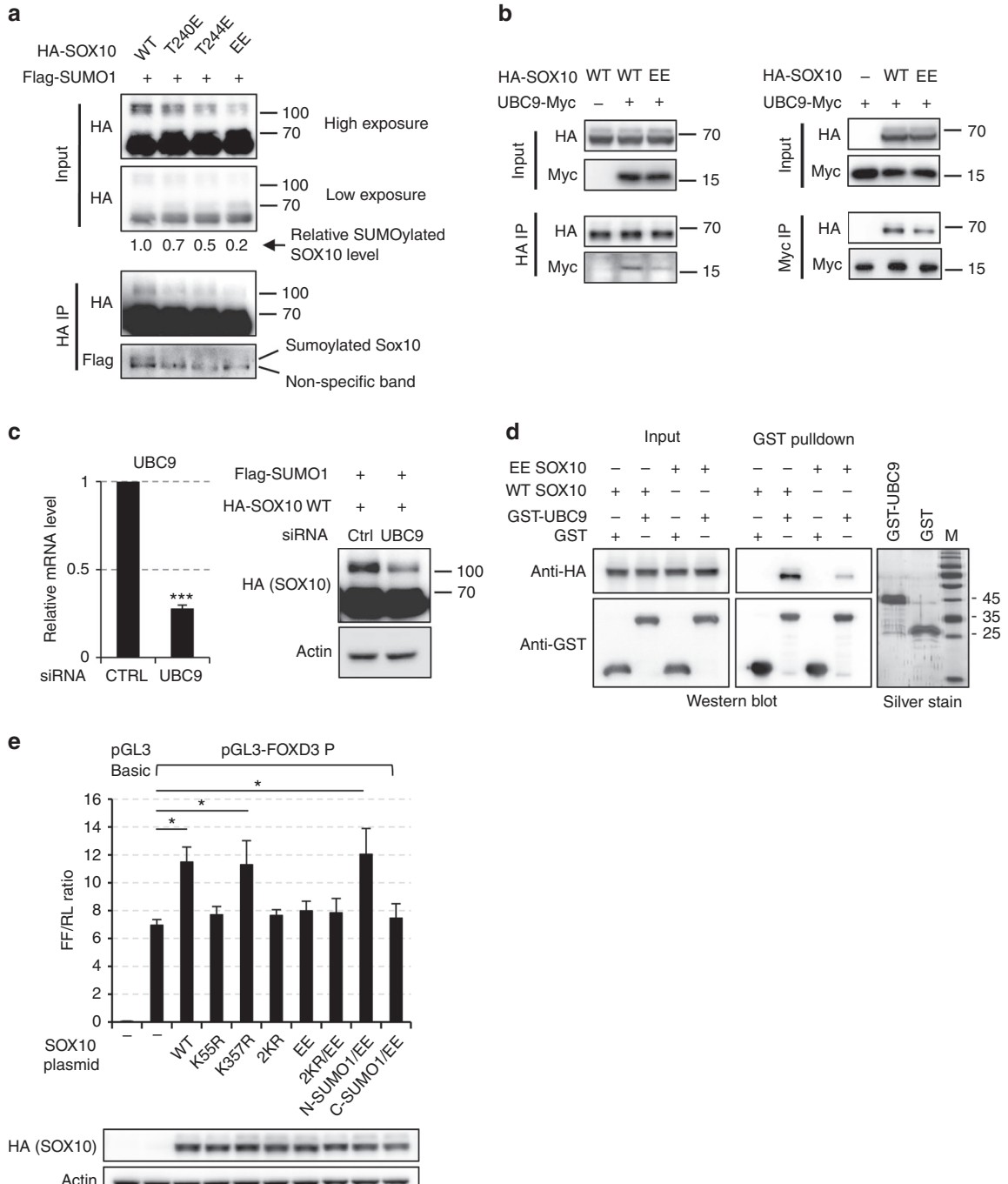

**Fig. 6** Phosphorylation at T240 and/or T244 inhibits SOX10 sumoylation. **a** HEK293T cells were co-transfected with plasmids expressing Flag-SUMO1 and one of the HA-SOX10 variants including WT, T240E, T244E, and EE. After 48 h, immunoprecipitation was performed with HA-tag antibody. Inputs and immunoprecipitates were analyzed by western blot. Relative levels of SOX10 sumoylation were quantitated by normalizing the intensities of the sumoylated bands against the non-sumoylated bands. **b** HEK293T cells were transfected with HA-SOX10 and UBC9-Myc expressing plasmids for 48 h. Reciprocal immunoprecipitations were performed using anti-Myc (left) or anti-HA (right) antibodies. Inputs and immunoprecipitates were analyzed by western blots. **c** HEK293T cells were co-transfected with WT HA-SOX10 and Flag-SUMO1 plasmids, ±UBC9 siRNAs for 48 h. Cells were then lysed for qPCR (left) and western blot (right) analysis. Average qPCR results from three independent experiments is shown and error bars represent standard deviation. Significance was determined by Student's two-tailed *t*-test, ***$p < 0.001$. **d** GST pull-down experiments were performed using recombinant GST-UBC9 or GST and sonicated lysates of 293T cells expressing WT or EE HA-SOX10. Input and pull-down proteins were analyzed by western blot. **e** HEK293T cells were co-transfected with 500 ng of pGL3-FOXD3, 50 ng of pRL-TK and 500 ng of indicated SOX10 plasmids. After 48 h, cells were lysed and dual-lucfiferase assays were performed. Average ratios of firefly and renilla luciferase activities (FF/RL) from three experiments are shown. The expressions of exogenous HA-SOX10 variants were verified by western blot. Error bars represent standard deviation. Significance was determined by ANOVA one-way test, *$p < 0.05$. Uncropped images are shown in Supplementary Fig. 14

observed in a FOXD3 promoter region proximal to the starting codon[31], raising the possibility that other regulatory elements may exist in this region. Indeed, we identify SOX10 as a novel regulator of FOXD3 in human mutant BRAF melanoma that binds to a conserved regulatory element located 270 bp upstream from the transcription starting site and activates FOXD3 transcription. Our findings are consistent with previous reports showing that SOX10 injection in Xenopus embryos led to enhanced expression of FOXD3 in cranial ganglia[22] and that ChIP-seq analysis detected SOX10 binding to FOXD3 locus in melanocytes[32]. Thus, in addition to the NC1 and NC2 enhancer-mediated regulation of FOXD3, the SOX10/FOXD3 axis discovered in mutant BRAF melanoma cells is likely a new component of the complex regulatory network that controls the development of neural crest.

Our work also describes a new mode of regulation of SOX10. ERK phosphorylation of SOX10 at T240 and/or T244 inhibits its transcription activity toward FOXD3 and other reported transcriptional targets, such as MITF, TYR, and SAMMSON (Fig. 4), indicating that this is a general regulation mechanism. However, it is not completely understood how phosphorylation of SOX10 at T240/T244 compromises its transcriptional activity. Blocking

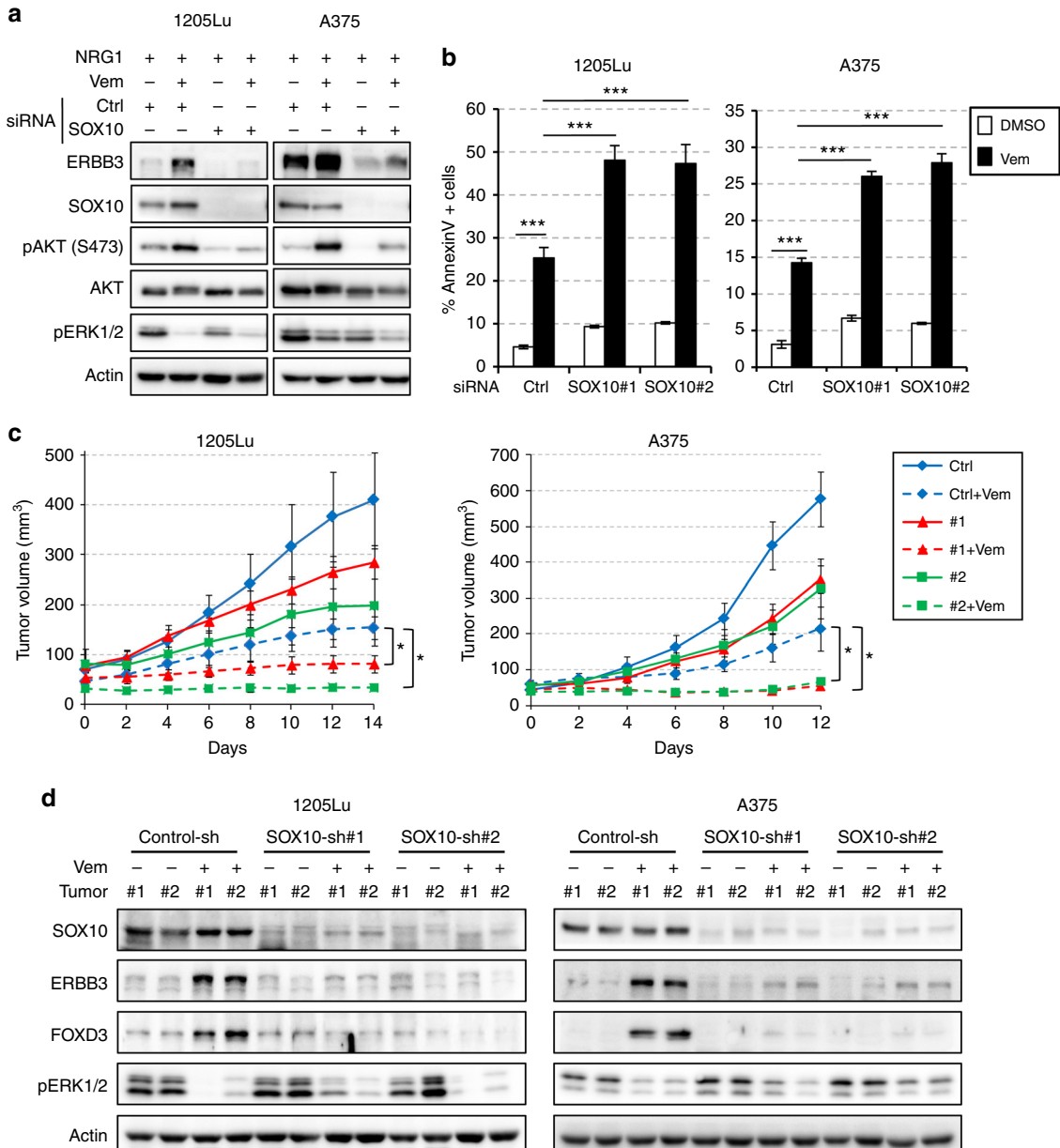

**Fig. 7** SOX10 depletion sensitizes melanoma cells to mutant BRAF inhibitor. **a** Melanoma cells were transfected with control or SOX10 #2 siRNAs for 72 h and ±2 μM vemurafenib for additional 24 h. Cells were then stimulated with 10 ng ml$^{-1}$ NRG1 for 1 h and lysed for western blot analysis. **b** Melanoma cells were transfected with Control, SOX10#1 or #2 siRNAs for 48 h and treated with ±10 μM Vemurafenib for additional 48 (1205Lu) or 72 (A375) hours. Cells were then collected and stained with Annexin-V/PI for flow cytometry analysis. Average percent of annexin-V positive cells from three independent experiments are shown. Error bars represent standard deviation. Significance was determined by ANOVA one-way test, ***$p < 0.001$. **c** Growth curves of tumors formed by 1205Lu-TR or A375-TR cells harboring control or SOX10-shRNAs in nude mice ($N = 7$ per condition). Statistical analyses (ANOVA test) were performed on tumor volume differences between RAFi-treated control-shRNA group and RAFi-treated SOX10-shRNA groups on day 12 (A375) or 14 (1205Lu). Error bars represent standard error. Significance was determined by ANOVA one-way test, *$p < 0.05$. **d** Western blot analysis of two sets of representative tumor samples excised on day 5. Uncropped images are shown in Supplementary Fig. 15

ERK signaling and hence SOX10 phosphorylation by RAF inhibitors does not alter nuclear localization or the DNA binding ability of SOX10 (Fig. 2e, f, Supplementary Fig. 4). Instead, we postulate that phosphorylation of SOX10 at T240/T244 might inhibit its transcription activity through interfering with SOX10 sumoylation based on three observations: (1) SOX10 is sumoylated at K55 and loss of this modification ablates its transcription activity; (2) SOX10 phosphomimetic mutants (T240E, T244E, and EE) showed reduced sumoylation levels compared with WT SOX10; (3). The SOX10 EE phosphomimetic mutant had decreased association with the E2 SUMO ligase, UBC9 and UBC9 knockdown led to reduced sumoylation of SOX10. This phosphorylation-sumoylation interplay is not unique to SOX10 and has also been reported in other proteins. Dependent on the cellular and protein contexts, phosphorylation of a protein can either facilitate or inhibit its sumoylation[33–36]. Our results of SOX10 represent another example of a mutually exclusive relationship between phosphorylation and sumoylation. While the phosphorylation-sumoylation interplay provides a reasonable mechanism for the regulation of SOX10 activity, it is still possible that phosphorylation of SOX10 exerts an inhibitory effect on its transcription activity by interacting with other transcriptional cofactors. Further investigation is needed to test these possibilities.

As an important mediator of adaptive resistance to RAF inhibitors, FOXD3 depletion promotes the cytotoxic effect of RAF inhibitors in mutant BRAF melanoma cells[12]. Consistently, we find that knockdown of SOX10, the upstream regulator of FOXD3 also sensitizes mutant BRAF melanoma cells to Vemurafenib in vitro and in vivo, suggesting that SOX10 can protect melanoma cells against the acute cytotoxic effect of RAF inhibitors. In addition, SOX10 knockdown by itself inhibited the growth of melanoma cells (Supplementary Fig. 7), a finding consistent with previous studies[17, 37]. Therefore, SOX10 not only exerts a cytoprotective role against RAF inhibitors, but also participates in the regulation of melanoma cell growth.

Our finding that SOX10 plays a protective role against acute RAF inhibitor treatment is seemingly contradictory to a previous report showing that loss of SOX10 contributes to RAF inhibitor resistance by activating TGF-β signaling and EGFR/PDGFRβ expression[18]. However, it should be noted that Sun et al. evaluated the effect of SOX10 depletion on RAF inhibitor resistance at a rather late time point of drug treatment (4 weeks, see Fig. 2e in Sun et al.[18]), when acquired resistance might have occurred. By contrast, our work focuses on the immediate (within 120 h) effects of SOX10 depletion on the survival of Vemurafenib-challenged cells, which is more pertinent to the time window of adaptive resistance. Therefore, the differential effects of SOX10 on drug resistance observed in these two works are likely due to different roles of SOX10 played at different stages of resistance development. At the initial stage of drug treatment, SOX10 may provide rapid protection on melanoma cells by upregulating pro-survival factors such as FOXD3, MITF, and SAMMSON and therefore is important for the survival of melanoma cells. In line with this, SOX10 expression is not reduced for at least 96–120 h of Vemurafenib treatment when FOXD3 is fully induced (Supplementary Figs. 1 and 2). However, when the treatment prolongs, other adaptive resistance mechanisms, for example, ERK reactivation, may be activated so that the apoptosis-protection burden of SOX10 is relieved and more benefits can be gained by gradual loss of SOX10, which as Sun's work suggested, may lead to the activation of the TGF-β signaling and EGFR/PDGFRβ expression.

Aside from its important roles in melanoma, SOX10 is also a key regulator of the neural crest development. Studies have shown that SOX10-dependent development of oligodendrocytes is inversely correlated with the activity of ERK or JNK kinase[38]. Thus, the phosphorylation-dependent regulation of SOX10 transcription activity in melanoma cells may have implication in the development field as well. Transgenic SOX10 EE knock-in mouse models will likely provide more insights into the role of this new mode of SOX10 regulation in the embryonic development.

## Methods

**Reagent**. Vemurafenib, SCH772984, and AZD6244 were purchased from Selleck Chemicals LLC (Houston, TX, USA). Doxycycline was purchased from Thermo Fisher Scientific (Rockford, IL, USA). Human Neuregulin-1 #5218 (hNRG-1) was purchased from Cell Signaling Technology (Beverley, MA, USA).

**Cell culture**. 1205Lu cells were gifted by Dr. Meenhard Herlyn at The Wistar Institute. M238 cells were gifted by Dr. Antoni Ribas at University of California, Los Angeles. A375 and HEK293T cells were purchased from the American Type Culture Collection (ATCC) (Manassas, VA, USA). 1205Lu-TR and A375-TR are sublines with high Tet repressor expression (Abel et al.[13]). 1205Lu, 1205Lu-TR, and M238 cells were cultured in RPMI 1640 medium with 10% fetal bovine serum and penicillin/streptomycin. A375, A375-TR, and HEK293T cells were cultured in DMEM medium with 10% fetal bovine serum and penicillin/streptomycin. Parental A375, 1205Lu, and M238 cells have been verified to carry the BRAF[V600E] mutation by sequencing. All cell lines were mycoplasma free.

**Western blotting**. Melanoma cell lysates were separated on SDS-PAGE gels and transferred to PVDF membranes. After blocking with 1% BSA for 1 h, the membranes were incubated with primary antibodies at 4 °C overnight. Next day, the membranes were incubated with horseradish peroxidase-conjugated secondary antibodies for 1 h at room temperature. Blots were then developed using an enhanced chemiluminescence western blotting detection kit (BioRad, Hercules, CA, USA). Antibodies against Phospho-p44/42 MAPK (Thr202/Tyr204, clone 197G2, #4377), FOXD3 (clone D20A9, #2019), HA-tag (clone 6E2, #2367, clone C29F4, #3724), Myc-tag (clone 71D10, #2278), HER3/ErbB3 (clone 1B2E, #4754), Phospho-Akt (Ser473, clone D9E, #4060), AKT (#9272), Phospho-MAPK Substrates Motif [PXpTP] (#14378) were purchased from Cell Signaling Technology (Beverley, MA, USA). Anti-β-actin (#A2066) and anti-FLAG-tag (clone M2, #F3165) were from Sigma-Aldrich. Anti-SOX10 (N-20, #SC-17342) was from Santa Cruz Biotechnology (Santa Cruz, CA, USA). Another anti-FOXD3 (#631702) antibody was purchased from Biolegend (San Diego, CA, USA).

**Quantitative RT-PCR**. Total RNA was extracted from melanoma cells by using the TriPure Isolation Reagent (Roche, Basel, Switzerland) and reverse transcribed into cDNA using iScript cDNA Synthesis Kit (BioRad, Hercules, CA, USA). PCR reactions were performed using iQ SYBR Green Supermix (BioRad) and analyzed by a CFX Connect real-time PCR detection system (BioRad). Relative mRNA levels were calculated using the comparative Ct (ΔCt) method. Each Quantitation of mRNA levels represents data from three independent experiments. The following primers were used: β-actin (forward, 5′-TACCTCATGAAGATCCTCACC-3′; reverse, 5′-TTTCG TGGATGCCACAGGAC-3′), FOXD3 (forward, 5′-CCCAA-GAACAGCCTAGTGAA-3′; reverse, 5′-GCAGTCGTTGAGTGAGAGGT-3′), MITF (forward, 5′-CCGTCTCTCACTGGATTGGT-3′; reverse, 5′-TACTTGGTGGGGTTTTCGAG-3′), TYR (forward, 5′-CAGCCCAGCAT-CATTCTTCTC-3′; reverse, 5′-GGATTACGCCGTAAAGGTCCCTC-3′), SAMM-SON (forward, 5′-CCTCTAGATGTGTAAGGGTAGT-3′; reverse, 5′-TTGAGTTGCATAGTTGAGGAA-3′).

**Dual-luciferase assay**. Around $3 \times 10^5$ HEK293T cells were transfected with pGL3-FOXD3 promoter constructs, HA-SOX10 expressing constructs and pRL-TK in 12-well plate using X-tremeGENE HP DNA transfection reagent (Roche). After 48 h, cells were collected for dual-luciferase assay using a Dual-Luciferase Reporter Assay Kit (Promega, Madison, WI, USA) according to manufacturer's instruction. Luminescence was detected by a FlexStations 3 microplate reader (Molecular Devices, Sunnyvale, CA, USA).

**Chromatin immunoprecipitation assay**. A375-TR HA-SOX10 WT and 1205Lu-TR HA-SOX10 WT cells were cultured in 15 cm dishes and treated with 100 ng mL$^{-1}$ doxycycline. After 72 h, cells were treated with or without 2 μM Vemurafenib for 6 h before lysed for ChIP analysis. Briefly, cells were fixed with 1% formaldehyde for 10 min and stopped with 0.125 M glycine. After wash by PBS, cells were scraped and collected by centrifugation. Cells were then resuspended in cell lysis buffer (20 mM Tris-HCL, pH 8.0, 85 mM KCL, 0.5% NP40, and protease inhibitors) and centrifuged to collect the nucleus. Nucleus pellet was lysed in SDS lysis buffer (1% SDS, 10 mM EDTA, 50 mM Tris-HCL, pH 8.1 and protease inhibitor) and sonicated to shear the DNA. Chromatin immunoprecipitation was then performed using diluted sonicated lysates (1:5 in dilution buffer, 0.01% SDS, 1.1% Triton X-100, 1.2 mM EDTA, 16.7 mM Tris-HCL, pH 8.1, 167 mM NaCl plus

protease inhibitors) and IgG or HA-tag antibody. Antibody-Chromatin complexes were captured by the protein A/G Plus-Agarose beads (Santa Cruz, CA, USA) and wished in low salt wash buffer (0.1% SDS, 1% Triton X-100, 2 mM EDTA, 20 mM Tris-HCL, pH 8.1, 500 mM NaCl), high salt wash buffer (0.1% SDS, 1% Triton X-100, 2 mM EDTA, 20 mM Tris-HCL, pH 8.1, 500 mM NaCl), LiCl wash buffer (0.25 M LiCl, 1% NP40, 1% deoxycholate, 1 mM EDTA, 10 mM Tris-HCl, pH 8.1) and TE Buffer (10 mM Tris-HCl, 1 mM EDTA, pH 8.0). Protein/DNA complexes were eluted with elution buffer (1% SDS, 0.1 M NaHCO$_3$) and decrosslinked in 0.2 M NaCl at 65 °C overnight. DNA was then purified by PCR cleanup columns. Immunoprecipitated chromatin DNA was detected by qPCR using iQ SYBR Green Supermix (BioRad). The following primers were used for PCR. NC_forward, 5'-ATGGTTGCCACTGGGGATCT-3'; NC_reverse, 5'-TGCCAAAGCCTAGGG-GAAGA-3'; FOXD3_forward, 5'-CACAGTGCGGAGCGGAGTT-3'; FOX-D3_reverse, 5'-ACGTGACCGTGCGTGAC-3'.

**Oligo pull-down assay.** A375-TR HA-SOX10 WT cells were treated with 100 ng mL$^{-1}$ doxycycline for 72 h to induce HA-SOX10 expression. After that, cells were treated with ±2 μM Vemurafenib for 6 h and lysed for nuclear extraction. Two 25-bp biotinylated sense and antisense oligos were annealed to form double-stranded DNA fragments containing the putative SOX10 binding site #3 or containing the mutated counterpart. The biotinylated, double-stranded DNA probe (10 μL of a 2.5 μM stock) and 10 μg poly dI/dC were added to 200 μg precleared nuclear extracts and incubated at 4 °C overnight. Non-biotinylated oligos of same sequence (NC) were used as a negative control. The following day, blocked streptavidin-agarose beads (NEB, Ipswich, MA, USA) were added to the lysate/DNA mixture and incubated for 2 h at 4 °C with rocking. After washing, the oligo pull-down samples were boiled in SDS lysis buffer and analyzed by western blot.

**In vitro kinase assay.** SOX10 peptides of WT sequence (HGPPTPPTTPKTELQ), T240A (HGPPAPPTTPKTELQ), T244A (HGPPTPPTAPKTELQ), or AA mutations (HGPPAPPTAPKTELQ) were synthesized using a CSBio 336X automated peptide synthesizer (CSBio, Menlo Park, CA, USA). For in vitro kinase assay, 0.5 mM peptide substrates were incubated with 200 unit recombinant ERK2 (NEB, Ipswich, MA, USA) and 0.5 mM ATP in 1X NEBuffer (NEB) at 30 °C for 45 min. The reaction products were analyzed by LC–Mass Spectrometry.

**In vivo detection of SOX10 phosphorylation.** A375-TR HA-SOX10 WT cells were treated with 100 ng mL$^{-1}$ doxycycline for 72 h and with ±2 μM vemurafenib for 6 h. Then cells were washed in PBS and lysed in lysis buffer (20 mM Tris-HCl, pH 7.5, 150 mM NaCl, 1 mM EDTA, 1% NP40, 0.1% SDS, 1% sodium deoxycholate) supplemented with protease and phosphatase inhibitor cocktails (Roche, Basel, Switzerland). HA-SOX10 was immunoprecipitated from the lysate using anti-HA Magnetic Beads (Thermo Fisher Scientific) and eluted with 50 mM NH$_4$HCO$_3$. Anti-HA immunoprecipitates were collected and digested in 50 mM NH$_4$HCO$_3$ with sequencing grade trypsin. The digested products were subsequently injected onto an AB SCIEX QTRAP 6500+ using Eksigent nanoflex cHiPLC system with a reverse-phase ChromXP C18-CL column for peptides separation at the flow rate of 300 nl min$^{-1}$. Peptides were eluted using a 62 min gradient from 95% solvent A (H$_2$O, 0.1% formic acid) and 5% B (acetonitrile, 0.1% formic acid) to 50% B in 41 min, 6 min at 90%B, and back to 5% for 10 min. The instrument was set to monitor 50 to 100 transitions in each sample with a dwelling time of 100 ms per transition. Eluted peptides were then electrosprayed into the mass spectrometer and MS/MS spectra were collected in the linear ion trap mode with a mass range of 100–1200[39]. The total ion chromatograms for the peptides eluted at identical time provided measurement of their relative quantities using Skyline software.

**Construction of lentiviral vectors and cell lines.** Wild-type HA-SOX10 cDNA was cloned into pENTR/D-TOPO vector (Thermo Fisher Scientific) to generate the entry plasmid. Entry plasmids of HA-SOX10 mutants were constructed using Quickchange site-directed mutagenesis kit (Agilent Technologies Inc., Santa Clara, CA, USA) and the WT HA-SOX10 entry plasmid as template. The resultant entry plasmids were recombined with pLentipuro/TO/V5-DEST to generate lentiviral plasmids. Lentiviruses were produced in HEK293FT cells and melanoma cells were infected with lentivirus for 72 h before selection with puromycin. For SOX10-shRNA constructs, DNA oligonucleotides were annealed and ligated into pENTR/H1/TO plasmid using the manufacturer's kit and instructions (Thermo Fisher Scientific). The shRNA targeting sequences are the same as the two SOX10 siRNAs. The shRNA cassettes were recombined into a destination vector with puromycin resistance. Lentiviruses were produced and melanoma cells were transduced as described above.

**Annexin V/PI apoptosis assay.** Cells were collected, washed by PBS and stained using the Annexin-V-FLUOS kit (Roche) according to manufacturer's protocol. Stained cells were analyzed by flow cytometry on a CytoFLEX system (Beckman Coulter, Indianapolis IN, USA). The data were analyzed using Flowjo software (Three Star, Inc., Ashland, OR, USA).

**siRNA transfection.** Melanoma cells were transfected with 12.5 nM small-interfering RNA and Lipofectamine RNAiMAX (Thermo Fisher Scientific) for 72 h. Non-targeting siRNA control (5'-UUCUCCGAACGUGUCACGU-3') and siRNAs for SOX10 (#1 5'-CCGUAUGCAGCACAAGAAA-3'; #2 5'-GUAUGCAGCA-CAAGAAAGA-3') were purchased from Shanghai GenePharma Co., Ltd. (Shanghai, China).

**GST pull-down experiments.** UBC9 cDNA was subcloned into pGEX-KG vector. E.coli BL21 cells harboring pGEX-UBC9 or pGEX-KG plasmids were grown to OD$_{600}$ = 0.5 and induced with 0.5 M isopropyl β--1-thiogalactopyranoside for 4 h. Cells were pelleted, resuspended in PBS supplemented with protease inhibitors and lysed by sonication. Recombinant proteins were purified using GST chromatography followed by a size exclusion chromatography on Superdex 75 column (GE health care, PA, USA). GST pull-down assays were carried out by incubating equal amounts of GST and GST-UBC9 immobilized on glutathione MagBeads (Gene-Script, NJ, USA) with lysates of 293T cells expressing WT or EE HA-SOX10, at 4 °C for 3 h. Protein/bead complexes were washed three times with washing buffer (20 mM Tris-HCl, pH 7.4, 300 mM NaCl, 0.5% NP40), eluted with SDS sample buffer and subjected to western blot analysis.

**Animal studies.** Five-week-old female BALB/c nude mice (Shanghai SLAC Laboratory Animal CO. LTD, Shanghai, China) were randomly divided into 6 treatment groups. 1205Lu-TR or A375-TR cells carrying Ctrl-shRNA, SOX10-shRNA #1 or SOX10-shRNA #2 were intradermally injected into mice, respectively, ($2 \times 10^6$ per mouse for 1205Lu and $4 \times 10^6$ per mouse for A375) and allowed to grow for 7–10 days to reach palpable tumor size (40–100 mm$^3$). The mice were then exposed to drinking water containing doxycycline (2 mg ml$^{-1}$) and treated intraperitoneally with Vemurafenib (30 mg kg$^{-1}$) or DMSO on a daily basis. Tumor sizes were measured every 2 days and tumor volumes were determined by the following formula: volume = (length × width$^2$) ×0.52. Sick mice or mice with their tumors damaged by cage mates were excluded from the experiment. Two mice from each treatment condition were killed on day 5 and tumors were excised for western blot analysis of the ERK/SOX10/FOXD3/ERBB3 signaling axis. The remaining mice were killed on day 12 (A375) or 14 (1205Lu). All animal protocols were approved by the Institutional Animal Care and Use Committee of Xi'an Jiaotong University. The investigators were not blinded to the experiment groups.

**Immunofluorescence assay.** 1205Lu-TR HA-SOX10 cells were cultured on coverslips in the presence of 100 ng mL$^{-1}$ Doxycycline for 72 h and then treated with 2 μM Vemurafenib for 0, 4, or 8 h. Cells were fixed in 3.7% formaldehyde for 15 min, and permeabilized in 0.1% Triton X-100 for 3 min. After that, cells were incubated with antibodies against HA tag at 4 °C overnight, followed by staining with TRITC-conjugated secondary antibody. Nucleus was stained by DAPI. Staining was visualized by an Inverted Microscope System (Nikon ECLIPSE Ti, Tokyo, Japan).

**Statistics.** Statistical significance of differences between the results was evaluated using Student's two-tailed t-test (assuming non-equal variance) for two groups' comparison or ANOVA one-way test where multiple groups were involved. A p-value <0.05 was considered statistically significant. Spearman's correlation was applied to estimate the correlation between expression of SOX10 and FOXD3 in mutant BRAF melanoma patients.

**Data availability.** All relevant data are available from the authors upon request.

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

## Acknowledgements

We thank Dr. Meenhard Herlyn (Wistar Institute, Philadelphia, PA, USA) and Dr. Antoni Ribas (University of California, Los Angeles, CA, USA) for kindly providing the 1205Lu and M238 melanoma cell lines respectively. Part of the gene expression correlation results in this work is based on RNA-seq data from the TCGA Research Network (http://cancergenome.nih.gov). This work was supported by the National Natural Science Foundation of China (31771557 to Y.S.), Fundamental Research Funds for Central Universities (xjj201489 to Y.S.), Natural Science Basic Research Plan in Shaanxi Province of China (2015JM8386 to Y.S.). Andrew E. Aplin and Timothy J. Purwin are supported by NIH R01 grants (CA160495 and CA196278) and the Dr. Miriam and Sheldon G. Adelson Medical Research Foundation. The National Basic Research Program (2015CB553602 to J. L.), the National Natural Science Foundation of China (31570777, 91649106 31770917 to J.L.) and Tianjin Applied Basic and Frontier Tech Major Project (12JCZDJC34400 to J.L.) and Tianjin Higher Education Sci-Tech Development Project (20112D05 to J.L.).

## Author contributions

Y.S. conceived and supervised the study. J.L., S.H., H.L., and Y.S. designed the experiments. S.H. performed most experiments with the help of Y.R., W.H., X.Z., Z.Z., B.M., Z. O., C.L., X.W., X.Q.W., H.Y., and Y.Z. T.J.P., T.Y., and R.Y. performed the bioinformatics study. J.L., S.H., and Y.S. analyzed the data. Y.S., J.L., S.H., and A.E.A. wrote the manuscript.

## Additional information

**Competing interests:** The authors declare no competing financial interests.

