## [Peer Review File(PDF 449 kb) · Nature Communications]

Reviewers' comments:

Reviewer #1 Expert in melanoma biology:

The manuscript by Han S et al. investigated the interplay between ERK1/2 and SOX10 in the context of BRAF-mutant melanoma. They showed that SOX10 is important for the adaptive activation of FOXD3 in BRAF-mutant melanoma cells treated with the BRAF inhibitor. This manuscript will be greatly strengthened by addressing the following questions -

1. In this study, the authors used vemurafenib alone. The combination of BRAFi plus MEKi becomes the standard care for treating patients with BRAF-mutant melanoma. How is the ERK-FOXD3-SOX10-ERBB3 signaling axis altered in BRAF-mutant melanoma cells treated with BRAFi plus MEKi?
2. Furthermore, will the authors be able to extend the in vitro studies to patients' tumor specimens?
3. In most cases, the authors treated cells up to 48 h. Will be ideal if the authors examine those cells when they are treated up to 72h-120h.
4. One can speculate that the ERK-FOXD3-SOX10-ERBB3 signaling axis might also plays a role in NRAS-mutant melanoma cells. Will the authors be able to treat NRAS-mutant melanoma cells with a specific MEK inhibitor?
5. Will the ERK-FOXD3-SOX10-ERBB3 signaling axis be correlated with patients' response to the targeted therapy as well as the overall survival and progression-free survival.

Reviewer #2 Expert in melanoma:

MAPK activation is a hallmark of melanoma. Targeted inhibition of this pathway, either through BRAF or MEK inhibitors is now standard of care for treating patients with metastatic melanoma. However, when used singly, or in combination, the majority of patients will relapse due to innate, acquired or adaptive drug resistance.

In this study Shao and colleagues investigate the mechanism by which the stemness factor FOXD3 is expressed following MAPK pathway inhibition. The transcriptional activity of SOX10 is regulated by sumoylation & phosphorylation. The authors show that K55 sumoylation is essential for its activity and that phosphorylation of T240 & T244 by ERK1/2 interferes with this process. They also show that SOX10 binds the FOXD3 promoter & is necessary & sufficient for expression of FOXD3 in BRAF mutant melanoma cells treated with MAPK pathway inhibitors, & that depletion of SOX10 sensitizes melanoma cells to MAPK inhibition – a potentially important therapeutic observation.

The authors have conducted a very thorough & meticulous functional assessment of SOX10 regulation of FOXD3 expression and the dependency of SOX10 activity on specific sumoylation and phosphorylation events. The methodologies are all highly appropriate and the experiments very carefully controlled, including some nice rescue experiments. Indeed, the experimental rigour exemplifies good molecular biology. The conclusions of the study are certainly supported by the data generated. The article is well written & clearly presented, and the story provides valuable information on the precise molecular interplay that governs one critical aspect of adaptive resistance to MAPK inhibition in melanoma. As such, it is a timely & welcome addition to the field.

Minor points to address:

Tables 1 & 2 should be combined.

Figure legends – change tense where needed in relation to what _is_ (not was) shown in the figures, e.g. replace “were shown” with “are shown”.

Figure 2A, C, E – the asterisks need to be larger & the 0 at the end of SOX10 in the first western blot panel of should not be wrapped to a second line.

Figure 4C – make asterisks larger.

Reviewer #3 Expert in SUMOylation:

Han et al have investigated the adaptive resistance process in mutant BRAF melanoma. They report that upregulation of FOXD3 upon ERK inhibition occurs through the transcription factor SOX10. They further propose that ERK-mediated phosphorylation reduces SOX10 SUMOylation, reducing its transcriptional activity. Finally, they demonstrate that SOX10 depletion sensitises mutant BRAF melanoma cells to RAF inhibitors.

The data presented are clear, convincing, well-performed, and the conclusions justified. The authors provide data that completes an elegant pathway into how mutant BRAF melanoma cells gain resistance to RAF inhibitors.

My primary concern with the data presented is whether the conceptual advance is sufficient to warrant publication in Nature Communications. While well-performed, many of the individual aspects of the story have been demonstrated previously – RAF inhibition is known to upregulate FOXD3, SOX10 has previously been demonstrated to be a SUMO substrate, and the concept of interplay between phosphorylation and SUMOylation in regulating substrate function (particularly that of transcription factors) is well-established.

I have a number of specific comments:

1. While the mass spec approach neatly demonstrates that T240 and T244 are phosphorylated in vivo and can be phosphorylated by ERK in vitro, I would like to see a more direct demonstration that ERK can phosphorylate SOX10 in cells. For example, by immunoprecipitation of HA-SOX10 WT or mutants of T240/T244 from control or cells treated with an ERK inhibitor followed by Western blotting for phospho-threonine.
2. Throughout the manuscript – does the SOX10 antibody used recognise the ‘rescue’ HA-SOX10? If so, in a number of cases, most notably Figure 4A, the SOX10 blot indicates the HA-SOX10 is under-rescuing, which could confound interpretation of the data. This is particularly relevant for Figure 4A, since under-rescue does not appear to be an issue in Figure 1C, which this figure is directly comparable to.
3. It would be good to show a lower exposure of the input from Figure 5B, to indicate the mutants are expressing at similar levels.
4. Similarly, a lower exposure for Figure 6A would confirm the mutants express to similar levels. Ideally, the authors would show quantification of SUMOylation of each of the mutants, since the effect here is quite subtle for the single mutants.
5. SOX10 phosphorylation could alter Ubc9 binding indirectly. The authors should confirm whether phospho-mimetic mutants show decreased binding to Ubc9 by using in vitro binding assays with purified proteins.
6. The authors should perform luciferase assays as in Figure 2A with non-SUMOylatable SOX10 to confirm a role for SUMOylation of SOX10 in driving expression of FOXD3. Furthermore, it would be good to analyse a non-SUMOylatable phospho-mimetic mutant, to directly test whether phosphorylation of SOX10 regulates FOXD3 transcription through altering SOX10 SUMOylation.
7. In the statistics section of the methods, the authors state that statistics were performed using

student's unpaired t-tests. In the vast majority of the manuscript, greater than two experimental groups are being compared. Surely ANOVAs would be more appropriate for comparing multiple groups? Reviewers' comments:

Reviewer #1 Expert in melanoma biology:

The manuscript by Han S et al. investigated the interplay between ERK1/2 and SOX10 in the context of BRAF-mutant melanoma. They showed that SOX10 is important for the adaptive activation of FOXD3 in BRAF-mutant melanoma cells treated with the BRAF inhibitor. This manuscript will be greatly strengthened by addressing the following questions -

1. In this study, the authors used vemurafenib alone. The combination of BRAFi plus MEKi becomes the standard care for treating patients with BRAF-mutant melanoma. How is the ERK-FOXD3-SOX10-ERBB3 signaling axis altered in BRAF-mutant melanoma cells treated with BRAFi plus MEKi?
2. Furthermore, will the authors be able to extend the in vitro studies to patients' tumor specimens?
3. In most cases, the authors treated cells up to 48 h. Will be ideal if the authors examine those cells when they are treated up to 72h-120h.
4. One can speculate that the ERK-FOXD3-SOX10-ERBB3 signaling axis might also plays a role in NRAS-mutant melanoma cells. Will the authors be able to treat NRAS-mutant melanoma cells with a specific MEK inhibitor?
5. Will the ERK-FOXD3-SOX10-ERBB3 signaling axis be correlated with patients' response to the targeted therapy as well as the overall survival and progression-free survival.

Reviewer #2 Expert in melanoma:

MAPK activation is a hallmark of melanoma. Targeted inhibition of this pathway, either through BRAF or MEK inhibitors is now standard of care for treating patients with metastatic melanoma. However, when used singly, or in combination, the majority of patients will relapse due to innate, acquired or adaptive drug resistance.

In this study Shao and colleagues investigate the mechanism by which the stemness factor FOXD3 is expressed following MAPK pathway inhibition. The transcriptional activity of SOX10 is regulated by sumoylation & phosphorylation. The authors show that K55 sumoylation is essential for its activity and that phosphorylation of T240 & T244 by ERK1/2 interferes with this process. They also show that SOX10 binds the FOXD3 promoter & is necessary & sufficient for expression of FOXD3 in BRAF mutant melanoma cells treated with MAPK pathway inhibitors, & that depletion of SOX10 sensitizes melanoma cells to MAPK inhibition – a potentially important therapeutic observation.

The authors have conducted a very thorough & meticulous functional assessment of SOX10 regulation of FOXD3 expression and the dependency of SOX10 activity on specific sumoylation and phosphorylation events. The methodologies are all highly appropriate and the experiments very carefully controlled, including some nice rescue experiments. Indeed, the experimental rigour exemplifies good molecular biology. The conclusions of the study are certainly supported by the data generated. The article is well written & clearly presented, and the story provides valuable information on the precise molecular interplay that governs one critical aspect of adaptive resistance to MAPK inhibition in melanoma. As such, it is a timely & welcome addition to the field.

Minor points to address:

Tables 1 & 2 should be combined.

Figure legends – change tense where needed in relation to what is (not was) shown in the figures, e.g. replace “were shown” with “are shown”.

Figure 2A, C, E – the asterisks need to be larger & the 0 at the end of SOX10 in the first western blot panel of should not be wrapped to a second line.

Figure 4C – make asterisks larger.

Reviewer #3 Expert in SUMOylation:

Han et al have investigated the adaptive resistance process in mutant BRAF melanoma. They report that upregulation of FOXD3 upon ERK inhibition occurs through the transcription factor SOX10. They further propose that ERK-mediated phosphorylation reduces SOX10 SUMOylation, reducing its transcriptional activity. Finally, they demonstrate that SOX10 depletion sensitises mutant BRAF melanoma cells to RAF inhibitors.

The data presented are clear, convincing, well-performed, and the conclusions justified. The authors provide data that completes an elegant pathway into how mutant BRAF melanoma cells gain resistance to RAF inhibitors.

My primary concern with the data presented is whether the conceptual advance is sufficient to warrant publication in Nature Communications. While well-performed, many of the individual aspects of the story have been demonstrated previously – RAF inhibition is known to upregulate FOXD3, SOX10 has previously been demonstrated to be a SUMO substrate, and the concept of interplay between phosphorylation and SUMOylation in regulating substrate function (particularly that of transcription factors) is well-established.

I have a number of specific comments:

1. While the mass spec approach neatly demonstrates that T240 and T244 are phosphorylated in vivo and can be phosphorylated by ERK in vitro, I would like to see a more direct demonstration that ERK can phosphorylate SOX10 in cells. For example, by immunoprecipitation of HA-SOX10 WT or mutants of T240/T244 from control or cells treated with an ERK inhibitor followed by Western blotting for phospho-threonine.
2. Throughout the manuscript – does the SOX10 antibody used recognise the ‘rescue’ HA-SOX10? If so, in a number of cases, most notably Figure 4A, the SOX10 blot indicates the HA-SOX10 is under-rescuing, which could confound interpretation of the data. This is particularly relevant for Figure 4A, since under-rescue does not appear to be an issue in Figure 1C, which this figure is directly comparable to.
3. It would be good to show a lower exposure of the input from Figure 5B, to indicate the mutants are expressing at similar levels.
4. Similarly, a lower exposure for Figure 6A would confirm the mutants express to similar levels. Ideally, the authors would show quantification of SUMOylation of each of the mutants, since the effect here is quite subtle for the single mutants.
5. SOX10 phosphorylation could alter Ubc9 binding indirectly. The authors should confirm whether phospho-mimetic mutants show decreased binding to Ubc9 by using in vitro binding assays with purified proteins.
6. The authors should perform luciferase assays as in Figure 2A with non-SUMOylatable SOX10 to confirm a role for SUMOylation of SOX10 in driving expression of FOXD3. Furthermore, it would be good to analyse a non-SUMOylatable phospho-mimetic mutant, to directly test whether

phosphorylation of SOX10 regulates FOXD3 transcription through altering SOX10 SUMOylation.

7. In the statistics section of the methods, the authors state that statistics were performed using student's unpaired t-tests. In the vast majority of the manuscript, greater than two experimental groups are being compared. Surely ANOVAs would be more appropriate for comparing multiple groups?

Response to Reviewers' comments

Reviewer #1 Expert in melanoma biology:

The manuscript by Han S et al. investigated the interplay between ERK1/2 and SOX10 in the context of BRAF-mutant melanoma. They showed that SOX10 is important for the adaptive activation of FOXD3 in BRAF-mutant melanoma cells treated with the BRAF inhibitor. This manuscript will be greatly strengthened by addressing the following questions

1. In this study, the authors used vemurafenib alone. The combination of BRAFi plus MEKi becomes the standard care for treating patients with BRAF-mutant melanoma. How is the ERK-SOX10-FOXD3-ERBB3 signaling axis altered in BRAF-mutant melanoma cells treated with BRAFi plus MEKi?

As the reviewer suggested, we investigated the ERK1/2-SOX10-FOXD3-ERBB3 signaling axis in mutant BRAF melanoma cells treated with a combination of Vemurafenib and Selumetinib (AZD6244, MEKi) over a period of 96 hours (**Supplementary Fig 2**). Similar to the results obtained with RAFi alone, combinational treatment of RAFi and MEKi effectively blocked the ERK1/2 signaling, induced the expression of FOXD3 and ERBB3, and activated the AKT signaling in response to NRG1 stimulation. Importantly, SOX10 depletion (with shRNA to enable long-term depletion) inhibited the upregulation of FOXD3/ERKBB3 and the activation of AKT signaling. Thus, the ERK1/2-SOX10-FOXD3-ERBB3 axis functions similarly in single RAFi- and RAFi/MEKi combo-treated mutant BRAF melanoma cells. (**Please refer to “Result” section, line 116-122, line 311-312 and Supplementary Fig 2**)

2. In most cases, the authors treated cells up to 48 h. Will be ideal if the authors examine those cells when they are treated up to 72h-120h.

To investigate whether the SOX10-dependent upregulation of FOXD3 by RAFi is durable, we extended the treatment time of Vemurafenib to 120 hours (**Supplementary Figure 1**). Our results showed that the induction of FOXD3 by RAFi is durable for at least 120 hours. Consistent with our prior observations, depletion of SOX10 expression (by treatment of doxycycline to induce SOX10 shRNAs) in A375 and 1205Lu cells effectively blocked the upregulation of FOXD3 by RAFi for up to 120 hours. In addition, the upregulation of ERBB3 and activation of AKT

signaling by inhibition of ERK1/2 signaling was also evident for up to 96 hours in Control-shRNA expressing cells, but was inhibited in SOX10-depleted cells (**Supplementary Figure 2 and reviewer #1, point 1**). Therefore, the ERK1/2-SOX10-FOXD3-ERBB3 axis is functional for at least 96-120 hours in mutant BRAF melanoma cells. (**Please refer to “Result” section, line 116-122, line 311-312, and Supplementary Figure 1-2**)

3. One can speculate that the ERK-SOX10-FOXD3-ERBB3 signaling axis might also plays a role in NRAS-mutant melanoma cells. Will the authors be able to treat NRAS-mutant melanoma cells with a specific MEK inhibitor?

According to a previous study (reference 10), induction of FOXD3 by inhibition of ERK1/2 signaling is specific to mutant BRAF melanoma cells. Melanocytes, WT BRAF melanoma cells or mutant BRAF thyroid cancer cells don't express FOXD3 upon MEKi treatment. To test whether FOXD3 can be induced in NRAS-mutant melanoma cells, a panel of NRAS-mutant melanoma cell lines were treated with the MEK inhibitor, AZD6244 (**Supplementary Fig. 3**). While MEKi effectively blocked ERK1/2 signaling in NRAS-mutant melanoma cells, no basal or induced FOXD3 expression was detected in these cells, suggesting that the ERK1/2-SOX10-FOXD3-ERBB3 signaling axis is specific to mutant BRAF melanoma cells. (**Please refer to “Result” section, line 119-122, and Supplementary Figure 3**)

4. Furthermore, will the authors be able to extend the in vitro studies to patients' tumor specimens? Will the ERK-SOX10-FOXD3-ERBB3 signaling axis be correlated with patients' response to the targeted therapy as well as the overall survival and progression-free survival.

We thank the reviewer for this constructive suggestion. While the alterations of the expression levels of SOX10 and FOXD3 in patients under Vemurafenib treatment have not been investigated, studies by Abel *et al.* have shown that levels of activated ERBB3 are significantly enhanced in some patients undergoing vemurafenib treatment (reference 12). Due to limited access to melanoma samples of treated patients, we are unable to analyze the ERK-SOX10-FOXD3-ERBB3 axis in patients at this stage. As an alternative, we established a mouse xenograft model to examine the *in vivo* relevance of the ERK-SOX10-FOXD3-ERBB3 signaling axis with tumor responses to RAFi. The *in vivo* results were highly consistent with our *in vitro* findings, showing that FOXD3 and ERBB3 were upregulated by RAFi and that SOX10 depletion not only blocked the induction of FOXD3 and ERBB3 but also sensitized melanoma tumors to RAFi (**Fig. 7d-e**). While our mouse xenograft model is not a perfect replacement of the clinical samples, we hope these new results can help convincing the audience on the *in vivo* relevance of our findings. (**Please refer to “Result” section, line 316-318, and Figure 7d-e**)

Reviewer #2 Expert in melanoma:

MAPK activation is a hallmark of melanoma. Targeted inhibition of this pathway, either through BRAF or MEK inhibitors is now standard of care for treating patients with metastatic melanoma. However, when used singly, or in combination, the majority of patients will relapse due to innate, acquired or adaptive drug resistance.

In this study Shao and colleagues investigate the mechanism by which the stemness factor FOXD3 is expressed following MAPK pathway inhibition. The transcriptional activity of SOX10 is regulated by sumoylation & phosphorylation. The authors show that K55 sumoylation is essential for its activity and that phosphorylation of T240 & T244 by ERK1/2 interferes with this process. They also show that SOX10 binds the FOXD3 promoter & is necessary & sufficient for expression of FOXD3 in BRAF mutant melanoma cells treated with MAPK pathway inhibitors, & that depletion of SOX10 sensitizes melanoma cells to MAPK inhibition – a potentially important therapeutic observation.

The authors have conducted a very thorough & meticulous functional assessment of SOX10 regulation of FOXD3 expression and the dependency of SOX10 activity on specific sumoylation and phosphorylation events. The methodologies are all highly appropriate and the experiments very carefully controlled, including some nice rescue experiments. Indeed, the experimental rigour exemplifies good molecular biology. The conclusions of the study are certainly supported by the data generated. The article is well written & clearly presented, and the story provides valuable information on the precise molecular interplay that governs one critical aspect of adaptive resistance to MAPK inhibition in melanoma. As such, it is a timely & welcome addition to the field.

Minor points to address:

1. *Tables 1 & 2 should be combined.*

We thank the reviewer for this suggestion. We have combined table 1 and 2 in the supplementary information. **(Please refer to “Supplementary table 1”)**

2. *Figure legends – change tense where needed in relation to what is (not was) shown in the figures, e.g. replace “were shown” with “are shown”.*

We thank the reviewer for pointing out these errors. We have made changes accordingly in the Figure legends. **(Please refer to “Figure legends” section)**

3. *Figure 2A, C, E – the asterisks need to be larger & the 0 at the end of SOX10 in the first western blot panel of should not be wrapped to a second line.*

We have enlarged the asterisks in Figure 2a, c, e and fixed the SOX10 label in the western blot. **(Please refer to Figure 2)**

4. *Figure 4C – make asterisks larger.*

We have enlarged the asterisks in Figure 4c. **(Please refer to Figure 4c)**

Reviewer #3 Expert in SUMOylation:

Han et al have investigated the adaptive resistance process in mutant BRAF melanoma. They report that upregulation of FOXD3 upon ERK inhibition occurs through the transcription factor SOX10. They further propose that ERK-mediated phosphorylation reduces SOX10 SUMOylation, reducing its transcriptional activity. Finally, they demonstrate that SOX10 depletion sensitises mutant BRAF melanoma cells to RAF inhibitors.

The data presented are clear, convincing, well-performed, and the conclusions justified. The authors provide data that completes an elegant pathway into how mutant BRAF melanoma cells gain resistance to RAF inhibitors.

1. My primary concern with the data presented is whether the conceptual advance is sufficient to warrant publication in Nature Communications. While well-performed, many of the individual aspects of the story have been demonstrated previously – RAF inhibition is known to upregulate FOXD3, SOX10 has previously been demonstrated to be a SUMO substrate, and the concept of interplay between phosphorylation and SUMOylation in regulating substrate function (particularly that of transcription factors) is well-established.

While we fully understand the concern of the reviewer on the conceptual advance of this work, we want to emphasize several contributions of this work to the fields of melanoma research and molecular biology:

1) RAFi-induced activation of the FOXD3/ERBB3/AKT axis is an important mechanism of adaptive resistance to RAF inhibitors in mutant BRAF melanoma. However, the mechanism of FOXD3 upregulation upon blockade of the ERK1/2 signaling remains unknown. Our work clarified the molecular mechanism of FOXD3 upregulation and depicted a complete signaling axis (ERK1/2-SOX10-FOXD3-ERBB3) that governs the adaptive resistance of melanoma cells to RAFi. These findings not only help us to understand the mechanism of adaptive resistance but also provide new therapeutic targets for melanoma treatment.

2) SOX10 is a pivotal transcription factor involved in the development of neural crest and the pathogenesis of several neural diseases and cancers. However, the knowledge on the regulation of SOX10 transcription activity is very limited. This work described a general mechanism for the regulation of SOX10 activity that involves an interplay between ERK-dependent phosphorylation of SOX10 at T240/T244 and sumoylation at K55. These findings expanded our knowledge on the regulation of SOX10 activity.

3) FOXD3 and SOX10 both play important regulatory roles during the development of neural crest; yet the regulatory relationship between the two and the chronological order in which they function during embryonic development is unknown. This work provided strong evidence for direct regulation of FOXD3 transcription by SOX10, which likely places SOX10 upstream of FOXD3 for functionality. These findings will help us better understanding the roles played by these two factors during the embryonic development.

Given the importance of SOX10 in many biological processes, we believe this work will be of great interest to a wide range of audience from the fields of melanoma research, molecular biology and neuroscience.

2. While the mass spec approach neatly demonstrates that T240 and T244 are phosphorylated in vivo and can be phosphorylated by ERK in vitro, I would like to see a more direct demonstration that ERK can phosphorylate SOX10 in cells. For example, by immunoprecipitation of HA-SOX10 WT or mutants of T240/T244 from control or cells treated with an ERK inhibitor followed by Western blotting for phospho-threonine.

We thank the reviewer for this thoughtful suggestion. As the reviewer suggested, we individually immunoprecipitated WT, T240A, T244A and AA HA-SOX10 variants from lentivirus transduced A375 cells treated with or without the ERK inhibitor, SCH772984 and probed phospho-threonine using an antibody targeting the PXPpTP motif (CST #14378). As expected, phospho-threonine was successfully detected in WT HA-SOX10 and the signal was reduced when cells were treated with ERKi (**Figure 3e**). Importantly, significantly less or no phospho-threonine signals were detected in T240A, T244A or AA HA-SOX10, confirming the phosphorylation of T240 and T244 sites *in vivo*. Moreover, the phosphorylation of SOX10 at T240/T244 was also observed in 293T cells expressing BRAF^{V600E} and was inhibited by MEK inhibitor (**Figure 3f**). These results suggested that T240 and T244 of SOX10 are phosphorylated in an ERK-dependent manner and these modifications are not cellular context specific. **(Please refer to “Result” section, line 200-209, and Figure 3e-f)**

3. Throughout the manuscript – does the SOX10 antibody used recognise the ‘rescue’ HA-SOX10? If so, in a number of cases, most notably Figure 4A, the SOX10 blot indicates the HA-SOX10 is under-rescuing, which could confound interpretation of the data. This is particularly relevant for Figure 4A, since under-rescue does not appear to be an issue in Figure 1C, which this figure is directly comparable to.

We thank the reviewer for raising this important concern. We agree that in the original Figure 4a, the expressions of exogenous HA-SOX10 appeared a bit lower than the endogenous ones. The variable expression levels of exogenous proteins are often caused by different titers of lentiviruses used to transduce the cells which vary batch to batch. To solve this problem, we regenerated 1205Lu/TR HA-SOX10 T240E, T244E and EE cell lines using a new batch of lentiviruses. These new cell lines now express exogenous HA-SOX10 at a comparable level to the endogenous ones (**Figure 4a**). Consistent with our prior observations, T240E, T244E or EE HA-SOX10 displayed compromised ability to induce FOXD3 expression upon RAFi treatment. **(Please refer to Figure 4a)**

4. It would be good to show a lower exposure of the input from Figure 5B, to indicate the mutants are expressing at similar levels.

We have rerun the samples in Figure 5b and showed the low and high exposures of the same

blot side by side. The low exposure blot indicated that WT and mutant HA-SOX10 were expressed at similar levels. Relative levels of sumoylated SOX10 were quantitated and shown in the figure. **(Please refer to Figure 5b)**

5. Similarly, a lower exposure for Figure 6A would confirm the mutants express to similar levels. Ideally, the authors would show quantification of SUMOylation of each of the mutants, since the effect here is quite subtle for the single mutants.

We have rerun the samples of Figure 6a and low/high exposures of the same blot were shown separately. Quantitation of sumoylated SOX10 was also shown in the figure. According to our results, the expression levels of SOX10 variants were at similar levels and the levels of sumoylated SOX10 were clearly reduced in phospho-mimetic mutants. **(Please refer to Figure 6a)**

6. SOX10 phosphorylation could alter Ubc9 binding indirectly. The authors should confirm whether phospho-mimetic mutants show decreased binding to Ubc9 by using *in vitro* binding assays with purified proteins.

As suggested by the reviewer, we initially attempted to purify recombinant WT His6x-SOX10, EE His6x-SOX10, GST-UBC9 and GST (as a negative control) proteins in *E. coli* for *in vitro* binding assays. While GST and GST-UBC9 were successfully expressed and purified, the His6x_SOX10 proteins turned out to be difficult to be expressed in *E. coli*. So we performed GST-pulldown experiments using the recombinant GST-UBC9/GST proteins and lysates of 293T cells transfected with WT or EE HA-SOX10 plasmids. The 293T lysates were sonicated before the pulldown assay to disrupt preexisting protein complexes. As shown in Figure 6d, WT SOX10 was efficiently pulled down by GST-UBC9 but not by the GST control, confirming the interaction between the SOX10 and UBC9. More importantly, EE SOX10 was pulled down less by GST-UBC9 when compared with WT SOX10. These results further confirmed that SOX10 interacts with UBC9 and that phosphorylation of SOX10 interferes with the binding between the two proteins. **(Please refer to “Result” section, line 275-281, and Figure 6d)**

7. The authors should perform luciferase assays as in Figure 2A with non-SUMOylatable SOX10 to confirm a role for SUMOylation of SOX10 in driving expression of FOXD3. Furthermore, it would be good to analyse a non-SUMOylatable phospho-mimetic mutant, to directly test whether phosphorylation of SOX10 regulates FOXD3 transcription through altering SOX10 SUMOylation.

We thank the reviewer for this constructive comment. As suggested, we performed dual luciferase assays to test the transactivation abilities of a panel of HA-SOX10 variants on FOXD3 promoter, including the sumoylation site mutants (K55R, K357R and 2KR HA-SOX10), phospho-mimetic mutant (EE HA-SOX10), non-SUMOylatable phospho-mimetic mutant (2KR/EE HA-SOX10) and two phospho-mimetic mutants that are either N-terminally or C-terminally fused to SUMO1 to mimick constitutive sumoylated SOX10 (C-SUMO1/EE, EE HA-SOX10 with C-terminal SUMO1 fusion and N-SUMO1/EE, EE HA-SOX10 with N-terminal SUMO1 fusion) **(Figure 6e)**. Taylor et al. (reference 21) have demonstrated that the C-terminal

SUMO1 fusion of SOX10 successfully recapitulated the function of sumoylated SOX10 at K357, a site close to the C-terminal end. In our system, we found that sumoylation of an N-terminal site K55 was more important than K357 for SOX10's transcription activity on FOXD3. Therefore, we constructed both N- and C-terminal SUMO1 fusions for comparison.

In accordance with our western blot results (Figure 5c-d), K55R and 2KR SOX10 failed to activate FOXD3 promoter while K357R SOX10 retained WT activity, confirming that K55 but not K357 is important for SOX10's transcription activity on FOXD3 promoter. The phospho-mimetic mutants EE and 2KR/EE SOX10 also lost their activities, consistent with our finding that T240/T244 phosphorylation compromise the transcription activity of SOX10 (Figure 4). Interestingly, we found that the N-terminal but not C-terminal SUMO1 fusion restored the transcription activity of EE SOX10 on FOXD3 promoter, supporting the idea that phosphorylation of SOX10 may regulate its transcription activity through altering SOX10 sumoylation at K55. **(Please refer to “Result” section, line 282-300, Figure 6e)**

8. In the statistics section of the methods, the authors state that statistics were performed using student's unpaired t-tests. In the vast majority of the manuscript, greater than two experimental groups are being compared. Surely ANOVAs would be more appropriate for comparing multiple groups?

We thank the reviewer for point this out and have reanalyzed the data using ANOVA test where appropriate. **(Please refer to “Methods” section, line 538-540)**

In addition to the above changes, we have also made corresponding revisions in “methods” and “figure legends” sections regarding the new experiments.

Reviewers' comments:

Reviewer #1 (Remarks to the Author):

This revised manuscript has been greatly improved. Specific comments are:

1. Fig 7d - can authors present the actual tumor volumes for this in vivo study?
2. Fig 7d - It seems that shSOX10 alone led to the substantial decrease in tumor volumes and this did not come up well in the manuscript. In fact, this piece of in vivo data was not in line with in vitro data as presented in Fig 7b and 7c because shSOX10 did not induce a significant amount of apoptotic cells.
3. Fig 7 - I think it is essential to carry out another in vivo study by treating A375 tumors in order to validate data based on 1205Lu tumors.

Reviewer #3 (Remarks to the Author):

Han et al have provided extensive new data to address the points raised in my original review. The new data are clear, convincing, and address each of my points fully. I now fully support publication of the manuscript in its current form, and compliment them on their elegant study.

Response to Reviewers' comments

Reviewer #1

This revised manuscript has been greatly improved. Specific comments are:

1. Fig 7d - can authors present the actual tumor volumes for this *in vivo* study?

As suggested by the reviewer, we replotted the *in vivo* xenograft results using the actual tumor volumes (see below and new Figure 7c). While both the old and new growth curves agreed on the conclusion that SOX10 depletion further improved the tumor-inhibiting effect of the RAF inhibitor *in vivo*, it looked more appropriate to present the growth curves using the actual tumor volumes in this case. We therefore took the review's advice and used the new plot in the revised manuscript. We really appreciate the reviewer for this wonderful suggestion.

2. Fig 7d - It seems that *shSOX10* alone led to the substantial decrease in tumor volumes and this did not come up well in the manuscript. In fact, this piece of *in vivo* data was not in line with *in vitro* data as presented in Fig 7b and 7c because *shSOX10* did not induce a significant amount of apoptotic cells.

We thank the reviewer for this comment. Indeed, down-regulation of SOX10 alone led to decreased tumor volumes *in vivo* (new Figure 7c). As the reviewer pointed out, SOX10 depletion alone only had marginal effects on melanoma cell apoptosis (Figure 7b). We therefore wondered whether loss of SOX10 expression may lead to growth inhibition. To test this, we performed MTT assay on 1205Lu and A375 cells with or without SOX10 knockdown. SOX10 depletion alone inhibited melanoma cell growth after 48-72 hours (See below or Supplementary Figure 7).

The regulation of melanoma cell proliferation by SOX10 has also been reported in other studies (see reference 16 and 36). Therefore, we believe that the reduced tumor volume observed in the SOX10-shRNA alone treated mice can be interpreted as a growth-inhibitory effect of SOX10 depletion. Based on these results, we conclude that SOX10 not only protects melanoma cells against the cytotoxicity of RAF inhibitors by triggering the SOX10/FOXD3/ERBB3/AKT axis-dependent adaptive resistance, but also contributes to the proliferation of melanoma cells.

References

16. Graf SA, Busch C, Bosserhoff AK, Besch R, Berking C. SOX10 promotes melanoma cell invasion by regulating melanoma inhibitory activity. *J Invest Dermatol.* 2014;134(8):2212-2220.
36. Cronin JC, Watkins-Chow DE, Incao A, Hasskamp JH, Schönewolf N, Aoude LG, Hayward NK, Bastian BC, Dummer R, Loftus SK, Pavan WJ. SOX10 ablation arrests cell cycle, induces senescence, and suppresses melanomagenesis. *Cancer Res.* 2013;73(18):5709-18.

3. Fig 7 - I think it is essential to carry out another *in vivo* study by treating A375 tumors in order to validate data based on 1205Lu tumors.

We have performed a second *in vivo* study using the A375 xenograft mouse model (new Figure 7c-d). Consistent with the results of the 1205Lu xenograft models, treatment of SOX10 shRNA alone reduced A375 tumor growth and more importantly, SOX10 knockdown further improved the tumor-inhibiting effect of Vemurafenib. This new *in vivo* study successfully validated our previous findings in 1205Lu cells and hopefully will strengthen our arguments on SOX10's role as a protective factor against RAFi *in vivo*. We thank the reviewer for this constructive suggestion.

We have made corresponding changes in “methods”, “figure legends” and “supplementary information” sections regarding the new experiments.

REVIEWERS' COMMENTS:

Reviewer #1 (Remarks to the Author):

The authors have successfully addressed all previous comments.